# The ubiquitin-conjugating enzyme UBE2QL1 coordinates lysophagy in response to endolysosomal damage

Lisa Koerver[1], Chrisovalantis Papadopoulos[1,†], Bin Liu[2,†] (iD), Bojana Kravic[1,†], Giulia Rota[1], Lukas Brecht[3], Tineke Veenendaal[4], Mira Polajnar[3], Anika Bluemke[1], Michael Ehrmann[1], Judith Klumperman[4], Marja Jäättelä[2] (iD), Christian Behrends[3,*] (iD) & Hemmo Meyer[1,**] (iD)

## Abstract

The autophagic clearance of damaged lysosomes by lysophagy involves extensive modification of the organelle with ubiquitin, but the underlying ubiquitination machinery is still poorly characterized. Here, we use an siRNA screening approach and identify human UBE2QL1 as a major regulator of lysosomal ubiquitination, lysophagy, and cell survival after lysosomal damage. UBE2QL1 translocates to permeabilized lysosomes where it associates with damage sensors, ubiquitination targets, and lysophagy effectors. UBE2QL1 knockdown reduces ubiquitination and accumulation of the critical autophagy receptor p62 and abrogates recruitment of the AAA-ATPase VCP/p97, which is essential for efficient lysophagy. Crucially, it affects association of LC3B with damaged lysosomes indicating that autophagosome formation was impaired. Already in unchallenged cells, depletion of UBE2QL1 leads to increased lysosomal damage, mTOR dissociation from lysosomes, and TFEB activation pointing to a role in lysosomal homeostasis. In line with this, mutation of the homologue ubc-25 in *Caenorhabditis elegans* exacerbates lysosome permeability in worms lacking the lysosome stabilizing protein SCAV-3/LIMP2. Thus, UBE2QL1 coordinates critical steps in the acute endolysosomal damage response and is essential for maintenance of lysosomal integrity.

**Keywords** autophagy; mTOR; p97; TAX1BP1; ubiquitin-conjugating enzyme
**Subject Categories** Autophagy & Cell Death; Post-translational Modifications & Proteolysis

See also: **N Mizushima** (October 2019)

## Introduction

Lysosomal membrane permeabilization (LMP) or rupture of late endosomes or lysosomes can be caused by many agents and conditions including lysosomotropic drugs, neurotoxic aggregates, pathogens, and alterations of lipid composition associated with aging or cancer. LMP can lead to the loss of lysosome integrity and lysosomal cell death [1]. Cells have therefore evolved sophisticated defense mechanisms, termed the endolysosomal damage response (ELDR) [2]. One element of this response is the repair of damaged lysosomes. This involves membrane stabilization by binding of the Hsp70 chaperone to certain lipids [3], and the $Ca^{2+}$-dependent rapid recruitment of ESCRT machinery that is thought to repair smaller holes in the lysosomal membrane [4,5]. A second branch concerns the activation of signaling cascades that triggers biogenesis of new lysosome components. It involves dissociation of the mTORC1 complex from lysosomes and, consequently, dephosphorylation of the transcription factor TFEB and its translocation into the nucleus [6,7]. Lastly, terminally defective lysosomes are cleared by a pathway of selective macroautophagy, termed lysophagy [8,9]. Lysophagy is initiated by the influx of cytosolic galectins, such as galectin-3 (Gal3, encoded by LGALS3) into the perforated lysosomes where they bind to β-galactosides on the luminal side of the membrane and most likely function as sensor of damaged lysosomes [10]. Gal3 binds the regulator TRIM16, and this is followed by extensive posttranslational modification of lysosomal proteins with ubiquitin [11,12].

One important role of ubiquitination is the recruitment of autophagy receptors such as p62/SQSTM1 that link the ubiquitinated organelle to LC3/Atg8 family members on autophagic membranes to promote phagophore formation [13,14]. p62 is critical for lysophagy, and its recruitment coincides with early detectable ubiquitination on the damaged lysosomes [11,15]. Another critical role of

1  Faculty of Biology, Centre for Medical Biotechnology, University of Duisburg-Essen, Essen, Germany
2  Cell Death and Metabolism Unit, Center for Autophagy, Recycling and Disease, Danish Cancer Society Research Center, Copenhagen, Denmark
3  Munich Cluster for Systems Neurology (SyNergy), Ludwig-Maximilians-Universität München, München, Germany
4  Section Cell Biology, Center for Molecular Medicine, University Medical Center Utrecht, Utrecht University, Utrecht, The Netherlands
   *Corresponding author. Tel: +49 89 4400 46509; E-mail: christian.behrends@mail03.med.uni-muenchen.de
   **Corresponding author. Tel: +49 201 183 4217; E-mail: hemmo.meyer@uni-due.de
   †These authors contributed equally to this work

ubiquitination is the recruitment of the ubiquitin-targeted AAA+-ATPase p97 (also called VCP). VCP/p97 is essential for efficient lysophagy [15,16] and is thought to extract factors that interfere with lysophagy, analogously to its function in mitophagy, the autophagic clearance of damaged mitochondria [2,16].

The general ubiquitination machinery that regulates lysophagy, consisting of an E1 ubiquitin activating enzyme along with specific E2 conjugating enzyme(s) and E3 ligase(s), is yet poorly characterized. The SCF ligase component FBXO27 has been shown to contribute to ubiquitination of damaged lysosomes and efficient lysophagy [17]. It is recruited through direct binding to the luminal glycosylation chains. However, the FBXO27 is only expressed in certain tissues and not present in model cells such as HeLa that display extensive lysosomal ubiquitination. In contrast to the roughly 600 E3 ligases, the human genome encodes for only about 40 E2 enzymes. Of note, while E3 ligases are required to fully activate E2s toward the substrates, the nature and reactivity of E2 enzymes can determine the length and linkage specificity of ubiquitin chains and thereby the ultimate fate of the substrate [18–20]. Moreover, E2 enzymes are considered as therapeutic targets because of growing evidence that mutations in E2s can modulate a multitude of diseases [21–23].

In this study, we therefore screened an siRNA library of E2 enzymes to identify regulators that mediate ubiquitination of lysosomal membrane proteins in response to damage induced by the lysosomotropic agent L-Leucyl-L-Leucine methyl ester (LLOMe) in HeLa cells. We identified a yet poorly characterized E2 ubiquitin-conjugating enzyme, UBE2QL1, and provide evidence that it is crucial for the damage-induced ubiquitination of lysosomes. Moreover, we show that UBE2QL1 coordinates the ordered recruitment of key regulators of lysophagy and is essential for efficient organelle clearance and cell survival after lysosomal damage. Finally, we provide evidence that UBE2QL1 has a conserved role in maintaining lysosomal integrity.

# Results

### A microscopy-based E2 siRNA screen identifies UBE2QL1 as a prime regulator of ubiquitination on damaged lysosomes

Ubiquitination of LAMP1-positive membranes upon LLOMe-induced damage can be robustly detected by the pan-ubiquitin antibody FK2 or antibodies specific for K48-linked ubiquitin chains with a high signal-to-noise ratio (Fig 1A). Using these readouts, we set up a three-step microscopy-based screen interrogating an siRNA library targeting 37 human E2s (Fig 1B). In the primary screen, HeLa cells were reverse-transfected with pools of four siRNAs targeting each E2. Cells were then treated with 250 µM LLOMe for 3 h to induce damage and processed for immunofluorescence. Automated high-content imaging with a spinning disk confocal microscope was followed by algorithm-based automated quantification. Based on colocalization of LAMP1 with FK2 or K48 signals, the percentage of ubiquitinated lysosomes and late endosomes was determined for each staining. Robust z-scores were calculated for each sample against the whole plate. Values of $\leq -2$ were considered a significant reduction in lysosome ubiquitination. Non-targeting siRNA did not affect ubiquitination, whereas control depletions such as of

ATG5 and ATG7 led to an increase of ubiquitinated lysosomes, as expected, due to inhibition of autophagic clearance (Fig 1C). Seven siRNA pools, targeting UBE2D1, UBE2E1, UBE2J1, UBE2D2, UBE2L6, UBE2Q2, and UBE2QL1, led to significant reduction in one or both ubiquitin stainings (Fig 1C, highlighted in red) and were therefore considered for a secondary screen. The result of the UBE2E2 pool was considered false positive, because experiments in an independent project revealed low depletion efficiency (data not shown).

In the secondary screen, pools were deconvolved into individual siRNAs, which were analyzed and evaluated as before. Individual siRNAs targeting three genes, UBE2J1, UBE2Q2, and UBE2QL1, scored for both FK2 and K48 stainings (Fig 1D, highlighted in red) and were therefore considered for further validation. In a third line of experiments, the respective cDNAs were expressed with HA-tag in HeLa cells to assess potential translocation to lysosomes upon damage (Fig 1E and F). UBE2J1 localized to the ER in control cells, as expected [24], and this did not change upon LLOMe treatment. UBE2Q2 and UBE2QL1 were diffusely distributed in untreated cells. Whereas UBE2Q2 localization was not affected by the LLOMe treatment, UBE2QL1 robustly translocated to damaged lysosomes. UBE2QL1 localization was specific to lysosomes, because it did not translocate to depolarized mitochondria (Fig EV1A) that are also extensively ubiquitinated for autophagic clearance [25]. As this suggested direct and specific involvement in lysosomal ubiquitination, UBE2QL1 was chosen for further analysis.

UBE2QL1 has only recently been described and its activity for ubiquitin been confirmed [26]. It is a member of the UBE2Q family of E2 enzymes but, in contrast to the other human members, UBE2Q1 and UBE2Q2, it lacks a RWD protein-interaction domain N-terminal of the catalytic UBC domain pointing to a functional specialization of UBE2QL1 within the family. Attempts to generate a CRISPR/CAS9-mediated knockout cell line failed (data not shown), suggesting that UBE2QL1 may be essential in HeLa cells. We raised an antibody against a C-terminal peptide that recognized a band at the expected size of roughly 18 kDa in HeLa cell lysate, whose identity was confirmed by siRNA-mediated depletion (Fig 1G). An siRNA targeting the open reading frame (oligo #5) depleted UBE2QL1 significantly (Fig 1G). Among the oligos from the screen, knockdown efficiency largely mirrored the observed effects in the screen. siRNA #4 that scored best depleted UBE2QL1 more efficiently than #2 (Fig 1G and Appendix Fig S1), whereas oligos #1 or #3 that did not score in the screen also reduced UBE2QL1 levels less if at all (Appendix Fig S1A). The antibody also detected a protein on lysosomes by immunofluorescence microscopy in LLOMe-treated but not in untreated cells (Fig EV1B and C). This signal was largely reduced by treatment with the three efficient siRNAs (#2, #4, and #5) demonstrating specificity (Fig EV1B and C). Thus, we confirm that also endogenous UBE2QL1 is recruited to damaged lysosomes.

We also tested whether UBE2QL1 responded to an alternative type of lysosomal damage. Neurotoxic aggregates such as tau amyloid fibrils can damage late endosomes/lysosomes when endocytosed from the extracellular space. We fed HeLa cells with Alexa-a488-labeled tau fibrils. Consistent with published data [15,27], some of the tau-containing LAMP1 vesicles were decorated with Gal3 indicating that the endocytosed fibrils damaged lysosomes (Fig EV1D). Of note, endogenous UBE2QL1 colocalized with individual of these damaged lysosomes suggesting that UBE2QL1 is

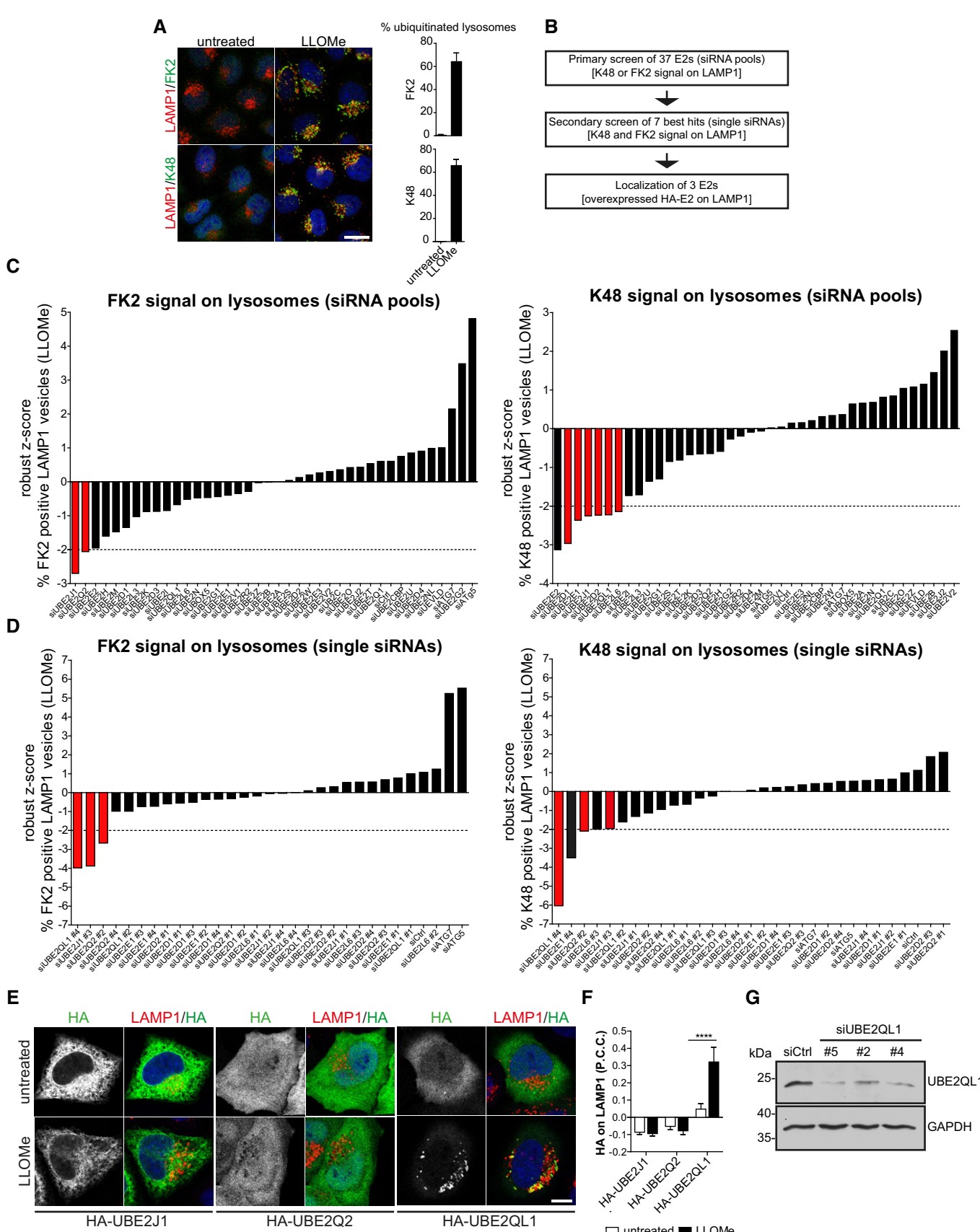

**Figure 1.**

◄

**Figure 1.  Microscopy-based siRNA screen of human E2 ubiquitin-conjugating enzymes identifies UBE2QL1 to be required for ubiquitination of damaged lysosomes.**

A  Robust immunofluorescence detection of ubiquitination on LAMP1-positive compartments with pan-ubiquitin FK2 or K48 chain-specific antibodies. HeLa cells were treated with 250 µM LLOMe or EtOH (untreated) for 3 h, fixed, and stained with indicated antibodies. Confocal microscopy was followed by automated image analysis. Graphs represent data from three independent experiments with ≥ 50 cells per condition (mean ± SD). Scale bar: 20 µm.

B  Screen overview.

C  Primary screen using pools of four single siRNAs targeting 37 human E2 enzymes. HeLa cells were seeded in 384-well plates, transfected with a siRNA library in quadruplicates for 72 h. Cells were LLOMe-treated and processed for imaging as in (A). Plates were imaged automatically with a spinning disk confocal microscope. Automated image analysis was done using the Acapella software, and the percentage of ubiquitinated lysosomes was determined. Robust z-scores were calculated for LLOMe-treated samples for each depletion against the values of the whole plate. Samples exceeding the threshold of −2 of robust z-score (highlighted in red) were considered for the secondary screen.

D  Selected candidates were re-screened with the same work pipeline as in (C) but with the four single siRNAs of pools of each candidate from the primary screen. Candidates exceeding the threshold in both stainings are highlighted in red.

E  Damage-induced translocation of UBE2QL1 to lysosomes. The three best candidates (from D) were transiently expressed with N-terminal HA-tag in HeLa cells. Cells were treated with LLOMe or EtOH (untreated) for 3 h, fixed, and stained for HA and LAMP1 as indicated and imaged by confocal laser scanning microscopy. Scale bar: 10 µm.

F  Automated quantification of (E). Shown are Pearson correlation coefficients (P.C.C) of HA and LAMP1 signals. Graph represents data from three independent experiments with ≥ 30 cells per condition (mean ± SD). ****$P < 0.0001$ (one-way ANOVA with Bonferroni's multiple comparison test).

G  Western blot detection of endogenous UBE2QL1 and depletion efficiency. Cells were transfected with control siRNA (Ctrl), two UBE2QL1 siRNAs from the screen (#2 or #4) or an additional siRNA (#5), and lysates probed with an antibody to UBE2QL1 as indicated. GAPDH was probed as loading control.

Source data are available online for this figure.

involved in the response to endomembrane damage caused by amyloid fibrils of tau.

## UBE2QL1 primarily regulates the delayed ubiquitination response on lysosomes

We next analyzed the effect of UBE2QL1 depletion on ubiquitination of damaged lysosomes in more detail in a single-well format. At least two types, K63 and K48-linked ubiquitin chains, are involved in lysophagy [11,15]. As expected, LLOMe treatment induced strong ubiquitination with both K48 and K63 chains as judged by immunofluorescence microscopy with chain-specific antibodies (Fig 2A–C). UBE2QL1 depletion with two independent siRNAs largely reduced the signal for K48 ubiquitin chains (Fig 2A and B). K63 chains were also reduced but not as extensively as K48 (Fig 2A and C). The reduced ubiquitination of damaged lysosomes was rescued by overexpression of wild-type UBE2QL1, but not of the catalytically inactive UBE2QL1(C88S) mutant (Fig EV2A–C), demonstrating that the effect of UBE2QL1 depletion was specific and not off-target. We next aimed to correlate the effect on ubiquitination with the localization of UBE2QL1. To do so, cells overexpressing HA-tagged UBE2QL1 were fixed at different time points after LLOMe-induced damage. The signals for K63 chains and p62 on lysosomes were apparent at 30 min and peaked within 1 h after LLOMe treatment (Fig 3A and B). UBE2QL1-HA was detected at these early time points, but only on few of the lysosomes. Instead, UBE2QL1 localization on lysosomes correlated with the slightly delayed appearance of K48 chains peaking at 2–3 h, when UBE2QL1 also colocalized with the majority of K48-decorated lysosomes (Fig 3A–C). Thus, whereas UBE2QL1-depletion also affects K63 ubiquitination, the bulk of UBE2QL1 coincides with the late K48-linked ubiquitination response.

## UBE2QL1 translocates in the vicinity of galectins, ubiquitination targets, and ubiquitin effectors at permeabilized lysosomes

To explore the localization of UBE2QL1 and K48-linked ubiquitin conjugates in more detail, we performed immuno-electron microscopy of LLOMe-treated cells. Because K48-chains were difficult to detect using this technique, we overexpressed a dominant-negative C160S mutant of the deubiquitinating enzyme GFP-YOD1 along with mCherry-Gal3 as damage marker. YOD1 cooperates with p97 in turning over K48-linked ubiquitin conjugates during lysophagy and becomes trapped with K48 chains when mutated [15]. Under these conditions, endogenous p62 was found on the outside of lysosomes, typically as a cap-like structure (Fig 4A), consistent with p62 recruiting the autophagosomal membranes. In contrast, lysosome-associated K48-ubiquitin chains and GFP-YOD1(C160S) were often found inside lysosomes together with mCherry-Gal3 (Figs 4B and EV3A and B). Confirming the immunofluorescence staining, overexpressed UBE2QL1-GFP localized to late endosomes and lysosomes and was often found in the lumen (Figs 4C and EV3C) in line with the localization of K48-linked ubiquitin conjugates.

To gain further insight into the role of UBE2QL1, we mapped the proteins in its vicinity by proximity biotinylation. UBE2QL1 was overexpressed with a C-terminal fusion of ascorbate peroxidase (APEX2), and biotinylation was induced in control and LLOMe-treated cells *in situ* by a pulse of biotin phenol and hydrogen peroxide. Biotinylated proteins from both conditions were compared by SILAC labeling and quantitative mass spectrometry. Four biological replica with high overlap of hits and correlation coefficients were evaluated (Fig 4D and E) and results summarized in a volcano plot (Fig 4F). Among the proteins largely increased in biotinylation after damage were lysosomal transmembrane proteins such as LIMP2 (also called SCARB2), NPC1, LAMP1, and LAMP2, although the latter just below the significance threshold. Of note, at least LAMP1 and LAMP2 become ubiquitinated upon damage [17], providing biochemical evidence that UBE2QL1 is recruited to lysosomes upon damage in vicinity of ubiquitination substrates. Interestingly, we also detected galectin-1 (LGALS1) and Gal3 (LGALS3) that bind to the glycans on the luminal side of the membrane of damaged lysosomes and that, along with galectin-8 (Gal8, LGALS8), are considered damage sensors [12]. This concurs with the EM data that UBE2QL1 can follow the galectins into permeabilized lysosomes and is consistent with the finding that also luminal parts of transmembrane proteins become ubiquitinated during lysophagy [17]. Of note, depletion of Gal3 or Gal8, or of both in

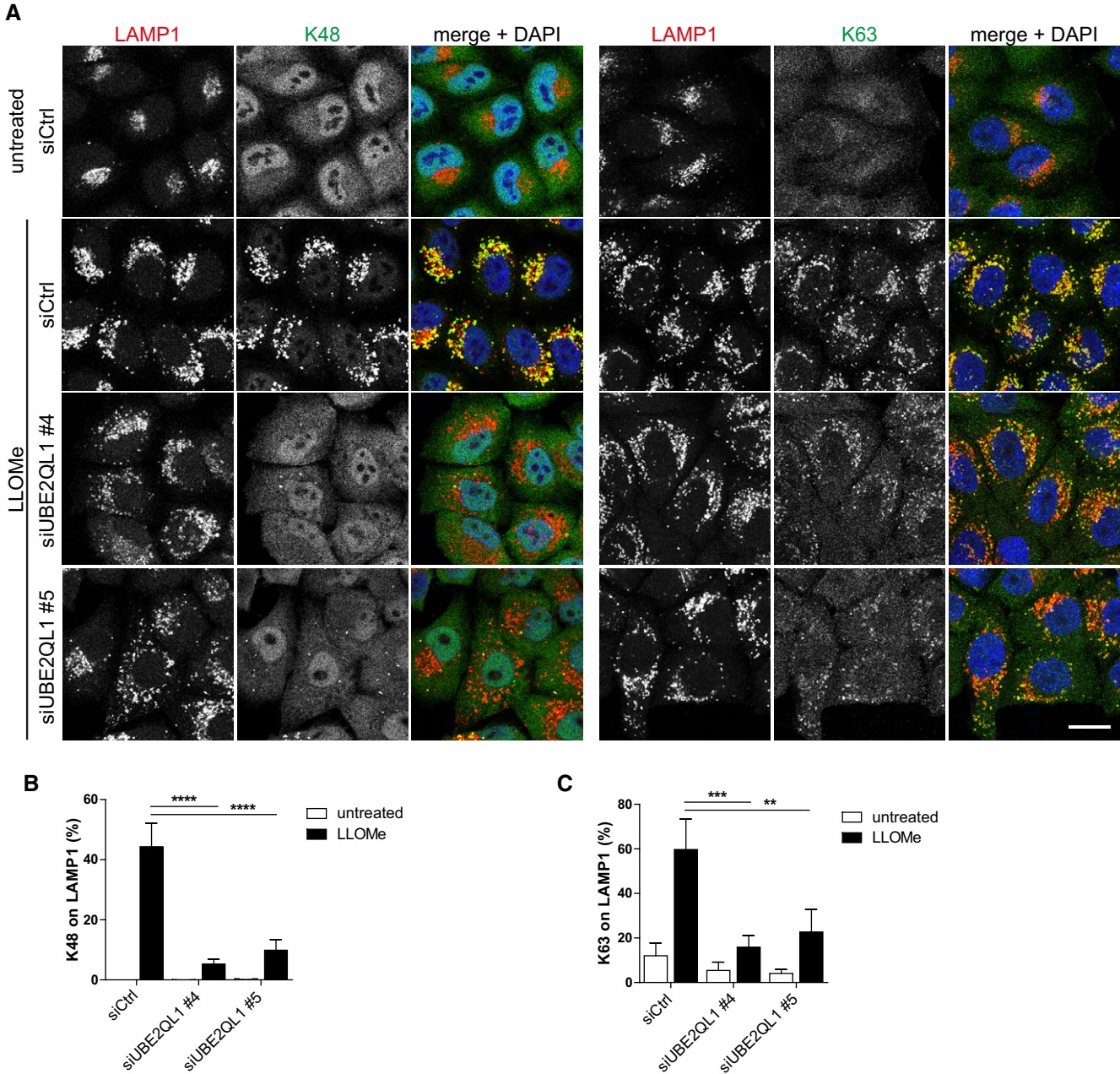

**Figure 2. UBE2QL1 depletion affects ubiquitination of damaged lysosomes.**

A   HeLa cells were transfected with different UBE2QL1 siRNAs for 60 h, and LLOMe or control treated (untreated) for 3 h. Cells were fixed and immunostained for LAMP1 along with antibodies specific for K63 or K48-linked ubiquitin chains as indicated and imaged with a laser scanning confocal microscope. Note the strong reduction of damage-induced K48 ubiquitination and decreased intensity of the K63 signal. Scale bar: 20 μm.

B, C   Automated image quantification of (A). Shown are percentages of K48 or K63-positive LAMP1 vesicles per cell. Graphs represent data from three independent experiments with ≥ 70 cells per condition (mean ± SD). **$P < 0.01$; ***$P < 0.001$; ****$P < 0.0001$ (one-way ANOVA with Bonferroni's multiple comparison test).

combination, did not affect translocation of UBE2QL1 to damaged lysosomes (Appendix Fig S2A–C), suggesting that UBE2QL1 has an independent recruitment path.

The assay also detected two autophagy receptors, TAX1BP1 and SQSTM1/p62 (the latter significantly increased, but below the stringent fold-change threshold of log2 H/L ≥ 1.5). Because TAX1BP1 has not been implicated in lysophagy before, we confirmed recruitment to damaged lysosomes by immunofluorescence microscopy (Fig EV3D). This suggests that UBE2QL1 functionally cooperates with both receptors. In addition, we robustly detected VCP/p97 and its cofactors including PLAA that target K48-linked ubiquitin conjugates [15]. Thus, these data provide evidence that UBE2QL1 translocates into the vicinity of key regulators of lysophagy at damaged lysosomes.

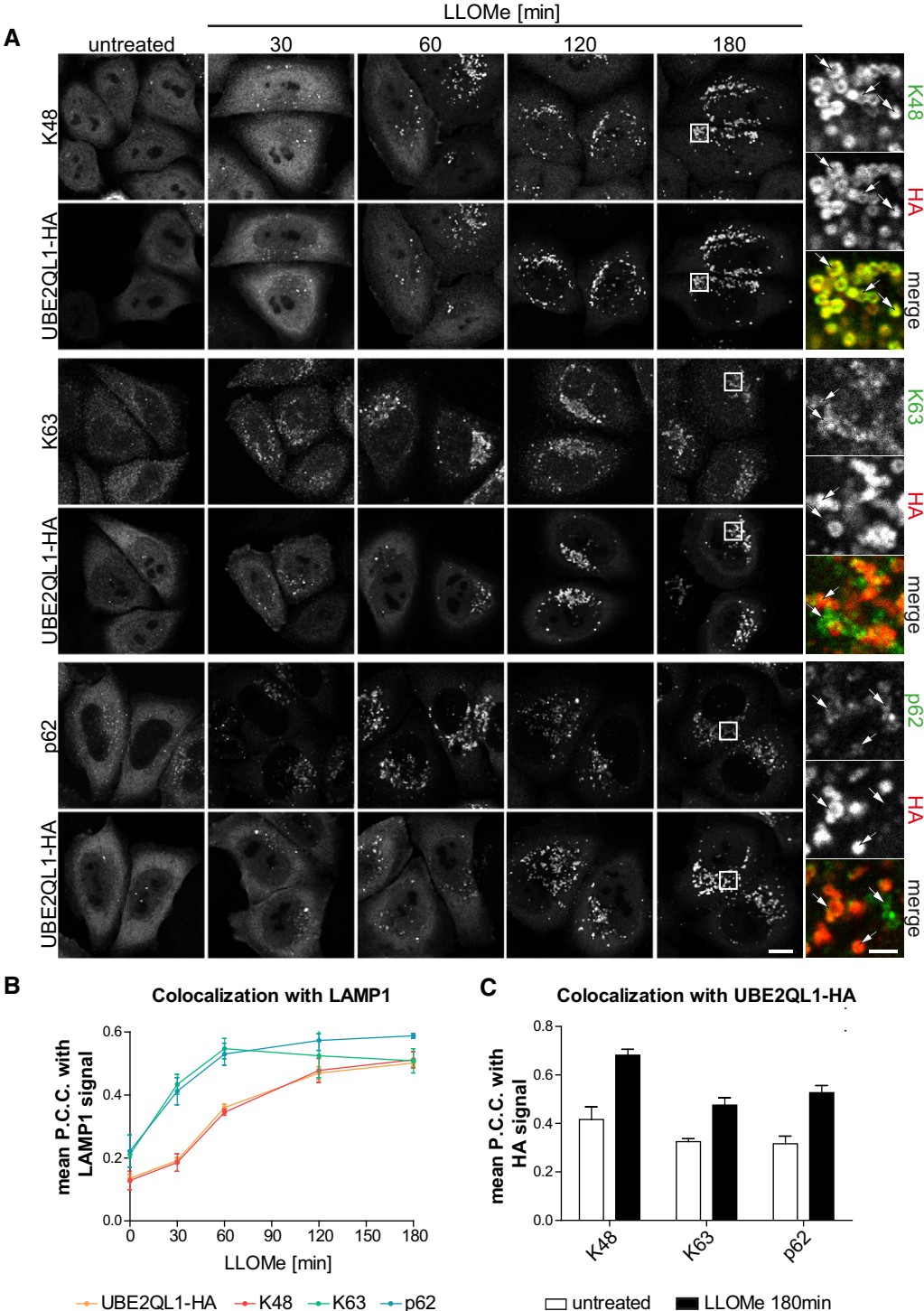

**Figure 3. UBE2QL1 recruitment coincides with the delayed K48-linked ubiquitination response.**

A   HeLa cells expressing UBE2QL1-HA were fixed at indicated time points after LLOMe treatment and processed for confocal microscopy. Cells were stained with antibodies specific for LAMP1 (not shown), and combinations of HA with K48 chains, K63 chains or p62 as indicated. Note the delayed co-emergence of K48 chains and UBE2QL1 vesicles, compared to the earlier appearance of K63 chains and p62. Arrows indicate colocalizing or non-colocalizing vesicles. Scale bars: 10 μm, 2 μm for inlays.

B   Automated quantification of (A). Shown are the Pearson correlation coefficients (P.C.C.) representing colocalization of UBE2QL1-HA, K48 chains, K63 chains, or p62 with LAMP1-positive vesicles. Graphs represent data from three independent experiments with ≥ 30 cells per condition (mean ± SD).

C   Automated quantification of (A). P.C.C.'s representing colocalization of K48 chains, K63 chains, or p62 with UBE2QL1-HA in cells treated with LLOMe or vehicle alone (untreated) for 180 min. Graphs represent data from three independent experiments with ≥ 30 cells per condition (mean ± SD).

## UBE2QL1 knockdown abrogates recruitment of VCP/p97, reduces accumulation of p62, and compromises association of LC3 with damaged lysosomes

Generally, p97 is recruited by ubiquitination of its target proteins. We therefore asked whether UBE2QL1-mediated ubiquitination underlies p97 recruitment to damaged lysosomes. We used stable p97-GFP expressing HeLa cells for convenient detection. Consistent with previous data [15], p97 was distributed in the cytosol in control cells but translocated to K48-decorated lysosomes upon treatment with LLOMe (Fig 5A and B). Depletion of UBE2QL1 with two independent siRNAs again largely reduced the signal for K48-chains (Figs 5A and EV4A). Crucially, this correlated with a dramatic decrease in the p97-GFP signal on damaged lysosomes

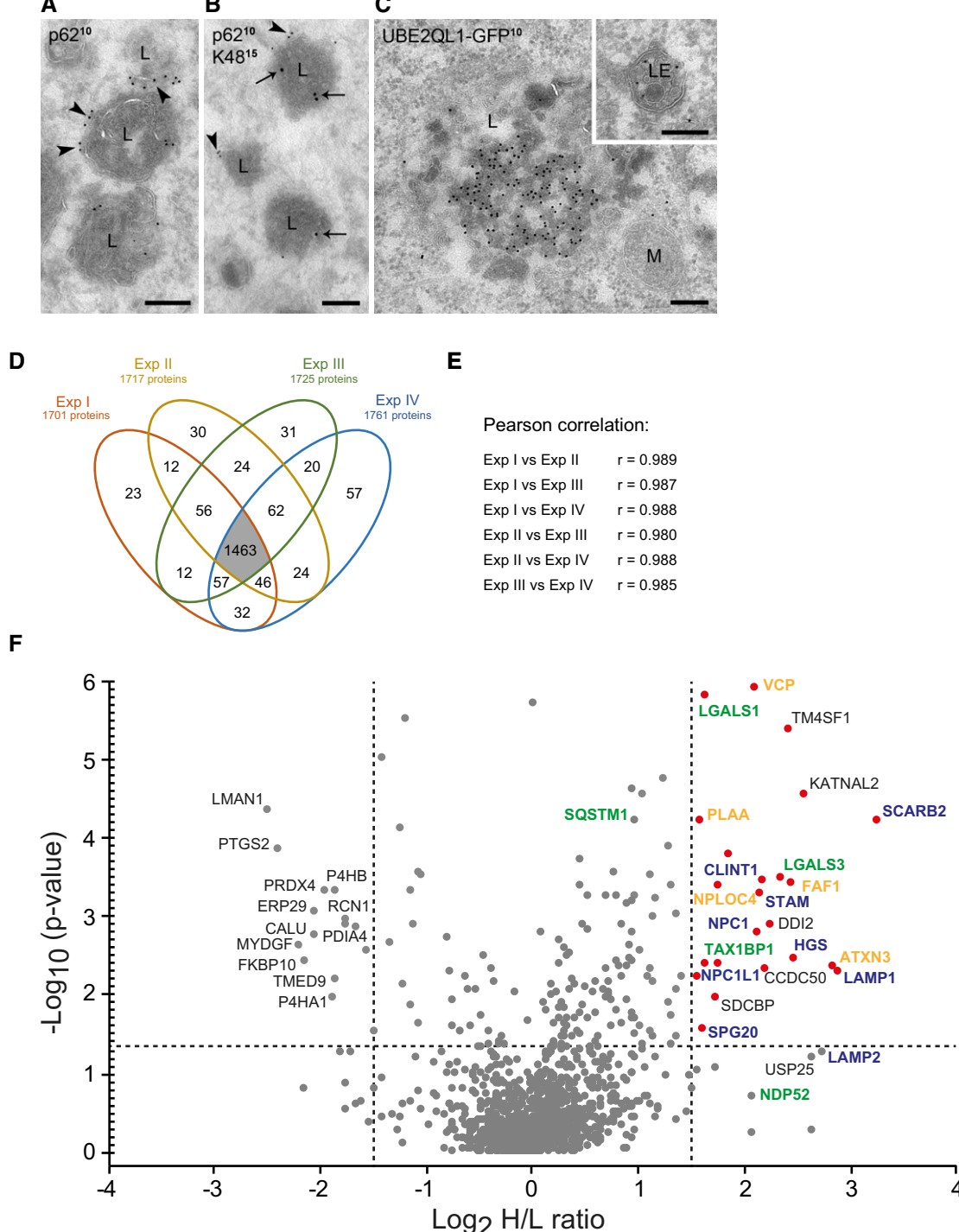

**Figure 4.**

◄

**Figure 4.  UBE2QL1 translocates in the vicinity of damage sensors, ubiquitination targets, and ubiquitin effectors at permeabilized lysosomes.**

A–C   Immuno-electron microscopy of LLOMe-treated HeLa cells. Cells in (A) and (B) overexpress mCherry-Gal3 and a dominant-negative C160S mutant of GFP-YOD1 and were immunostained for p62 and K48. (A) p62 (10 nm gold particles) localizes as a cap (arrowheads) to the outer limiting membrane of lysosomes (L). (B) K48 ubiquitin chains (15 nm gold particles, arrows) are found in the lumen of lysosomes that also contain p62 (10 nm gold particles, arrowheads) at their outer membrane. (C) Overexpressed UBE2QL1-GFP (10 nm gold particles) is found in late endosomes (LE, Inset) and lysosomes. M = mitochondrion. Scale bars: 200 nm.

D–F   Comparative UBE2QL1 proximity mapping before and after lysosome damage using SILAC mass spectrometry. HeLa cells expressing a UBE2QL1-APEX2 fusion were EtOH (light-labeled, L)- or LLOMe (heavy-labeled, H)-treated and pulsed with biotin phenol (30 min) and $H_2O_2$ (1 min). Proteins detected in replicative streptavidin purifications are depicted in a Venn diagram (D) and displayed high correlation coefficients (E). (F) Results from the four experiments are summarized in a volcano plot. Horizontal dotted line represents the significance threshold ($P > 0.05$). The vertical lines indicate the fold-change cutoff ($\log 2$ (H/L) $\geq$ 1.5). Red dots indicate proteins above the significance and fold-change thresholds. Protein names are color-coded in blue (endosome/lysosome-associated), green (endolysosomal damage/autophagy-associated), and orange (VCP/p97 and its cofactors).

Source data are available online for this figure.

(Figs 5A and B, and EV4A), demonstrating that UBE2QL1-regulated ubiquitination drives the translocation of p97 to damaged lysosomes and recruitment to its targets. Moreover, also p62 was reduced in UBE2QL1-depleted cells compared to control cells treated with LLOMe (Figs 5C and D, and EV4B).

Both factors, p62 and p97, are essential for efficient formation of LC3-decorated phagophores around damaged lysosomes [15]. We therefore tested localization of LC3B and co-stained for Gal3 to distinguish damaged lysosomes from LC3B-containing autolysosomes. In control cells, many Gal3-decorated damaged lysosomes were associated with LC3B, as expected, whereas they largely lacked LC3B in UBE2QL1-depleted cells (Fig 5E and F). Western blot analysis confirmed that this was not due to decreased formation of the lipidated LC3-II form upon LLOMe treatment and that UBE2QL1 depletion did not generally affect autophagy flux (Fig 5G). Thus, UBE2QL1-regulated ubiquitination is essential for recruitment of critical autophagy effectors and for efficient formation of autophagosomes to engulf damaged lysosomes.

## UBE2QL1 is essential for efficient clearance of damaged lysosomes and cell survival

Given the strong effects on autophagosome formation, we asked whether UBE2QL1 is required for efficient clearance of damaged lysosomes by lysophagy. To address this question, we performed the Gal3 puncta assay [10]. As expected, LLOMe treatment induced translocation of Gal3 from the cytosol to lysosomes, and these Gal3-decorated lysosomes were largely cleared in the course of 10 h in control cells (Fig 6A and B). Importantly, the clearance of LLOMe-induced Gal3 puncta was compromised in UBE2QL1-depleted cells resulting in persisting Gal3-decorated lysosomes even at 10 h after LLOMe treatment (Fig 6A and B). Of note, we detected Gal3-positive puncta also before LLOMe treatment in cells depleted with oligo #5 (Fig 6A and B), and after extended incubation also with oligo #4 (Appendix Fig S3A) indicating that UBE2QL1 has a constitutive role in maintaining lysosome integrity even in otherwise unchallenged cells. Furthermore, the Gal3 signal after LLOMe treatment was much more pronounced in UBE2QL1-depleted cells than in control cells (Fig 6A). This effect in unchallenged cells was not observed in ATG5- or ATG7-depleted cells (Appendix Fig S3B and C). We next challenged control and UBE2QL1-depleted cells with increasing LLOMe concentrations for 12 h and monitored cell viability by MTS assays. Consistent with a role of UBE2QL1 in promoting efficient lysophagy, cell viability was significantly reduced in UBE2QL1-depleted compared to control-depleted cells (Fig 6C).

Lysosomal membrane permeabilization can also be induced by functional inhibitors of the acidic sphingomyelinase such as terfenadine that lead to accumulation of sphingomyelin in the lysosomal membrane and thereby destabilize the membrane [28]. Using Gal3 as a marker, we confirmed the damage induced by terfenadine treatment with many Gal3 puncta emerging compared to control-treated cells (Appendix Fig S4). Importantly, the number and size of damaged lysosomes dramatically increased in UBE2QL1-depleted cells (Appendix Fig S4). Thus, UBEQL1 is essential for the response to different types of damaging agents.

## UBE2QL1 and its homologue UBC-25 in *Caenorhabditis elegans* are essential for maintenance of lysosomal integrity

The increased number of damaged lysosomes observed in UBE2QL1-depleted but otherwise unchallenged cells (Fig 6A and B) indicated a constitutive function of UBE2QL1 in lysosomal homeostasis. Compromised lysosomal integrity can lead to dissociation of mTORC1 from lysosomes and dephosphorylation of the transcription factor TFEB that then translocates from the cytosol to the nucleus to induce biogenesis of lysosomal components [7]. Indeed, TFEB displayed increased nuclear localization in cells treated with two different UBE2QL1 siRNAs compared to control cells (Figs 7A and B, and EV5A). In line with this, mTOR partially dissociated from lysosomes (Figs 7C and D, and EV5B) indicating that lysosomal stress signaling was activated in UBE2QL1-depleted cells, and the number of LAMP1-positive vesicles was increased (Fig 7E). Western blot analysis confirmed these findings with an increase of LAMP1 protein, dephosphorylation of TFEB, and decreased phosphorylation of the ribosomal protein S6, indicating reduced mTOR activity (Fig 7F). These changes in UBE2QL1-depleted cells were partially rescued by overexpression of wild-type UBE2QL1, but not of the catalytic mutant UBE2QL1(C88S) (Fig EV5C).

We next asked whether the function of UBE2QL1 in maintaining lysosomal homeostasis was conserved in *C. elegans*. The nematode is a useful genetic model for lysosomal integrity that can conveniently be monitored by transgenic expression of GFP-Gal3 [10]. Confirming published data on the role of SCAV-3, the orthologue of human LIMP2 [29], lysosomes were largely destabilized in animals harboring the *scav-3(qx193)* loss-of-function allele compared to wild-type (N2) as evidenced by the emergence of Gal3-positive membrane compartments in hypodermal hyp7 cells of worms at the L4 + 24 h stage (Fig 7G–I). *Caenorhabditis elegans* has only one member of the UBE2Q family, UBC-25. Importantly, lysosomal permeability in *scav-3(qx193); ubc-25(ok1732)* double mutant

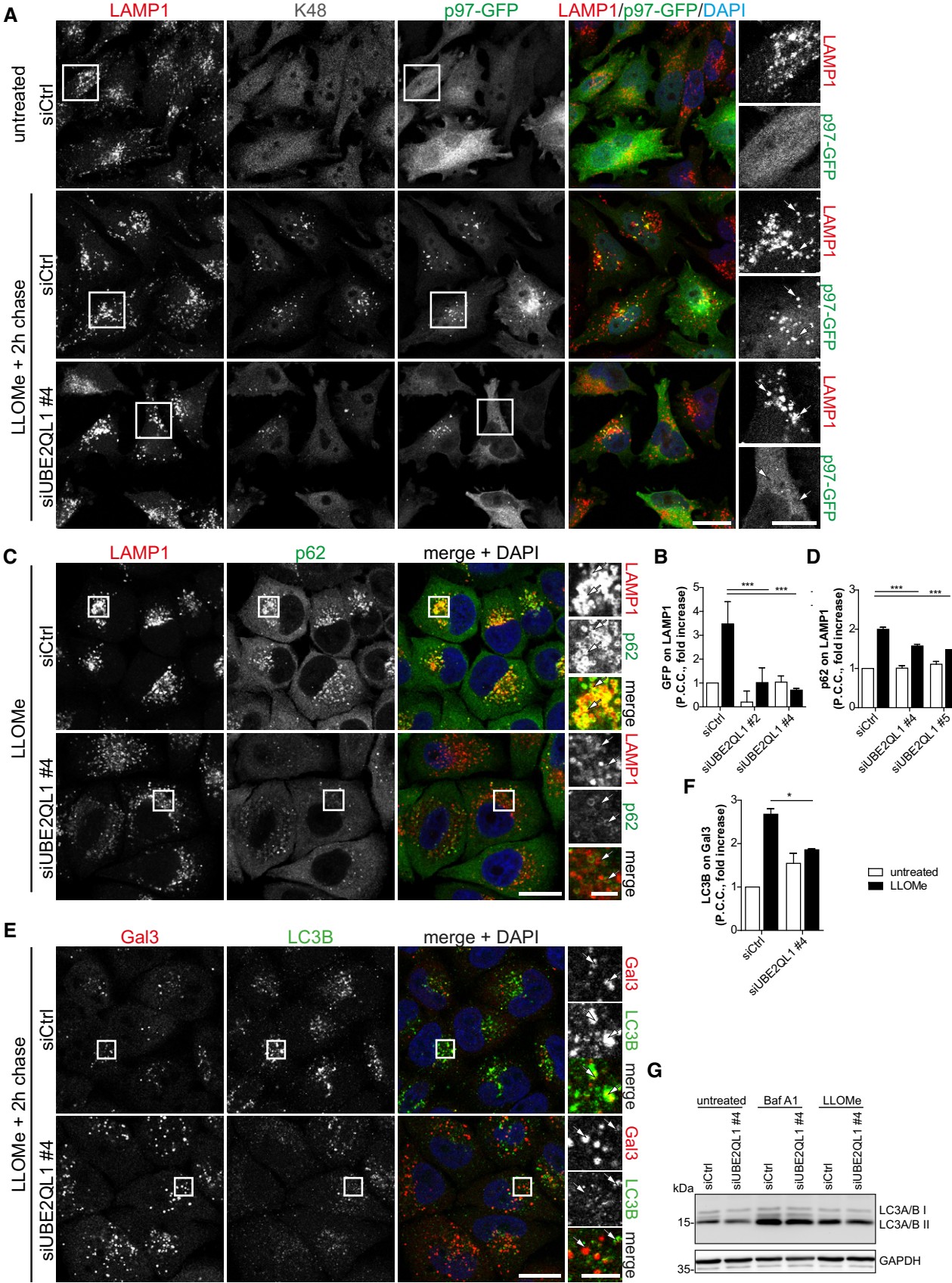

**Figure 5.**

**Figure 5.  UBE2QL1 abrogates recruitment of VCP/p97, reduces accumulation of p62, and compromises association of LC3 with damaged lysosomes.**

A   HeLa cells stably expressing p97-GFP were control or UBE2QL1-depleted for 48 h, and expression of p97-GFP was induced for the last 24 h. Cells were LLOMe or control treated for 1 h, fixed after a chase of 2 h, and stained with antibodies to K48 ubiquitin chains and LAMP1 as indicated. Note the reduction of the K48 and p97-GFP signals on damaged lysosomes in UBE2QL1-depleted cells. Arrows indicate colocalizing or non-colocalizing vesicles. Scale bars: 20 μm, 10 μm for inlays. See Fig EV4A for images of cells treated with an alternative siRNA.

B   Automated quantification of (A) and Fig EV4A. Shown are fold increases of Pearson correlation coefficients (P.C.C.) of p97-GFP and LAMP1 signals, normalized to untreated siCtrl. Data are from four independent experiments with ≥ 30 cells per condition.

C   HeLa cells were control or UBE2QL1-depleted for 60 h, fixed after 1 h of LLOMe or control treatment and stained for LAMP1 and p62, and analyzed by confocal microscopy. Arrows indicate colocalizing or non-colocalizing vesicles. Scale bars: 20 μm, 5 μm for inlays. See Fig EV4B for images of cells treated with an alternative siRNA and control cells.

D   Quantification of (C) and Fig EV4B. Shown are fold increases of Pearson correlation coefficients (P.C.C.) of p62 on LAMP1 signals, normalized to untreated siCtrl. Graph represents data from three independent experiments with ≥ 30 cells per condition.

E   HeLa cells were treated as in (C), but fixed after additional 2 h of chase. Samples were stained for galectin-3 (Gal3) to detect damaged lysosomes, and LC3B as marker for autophagosomes. Note the reduction of LC3B colocalization with Gal3 indicating impaired autophagosome formation. Arrows indicate colocalizing or non-colocalizing vesicles. Scale bars: 20 μm, 5 μm for inlays.

F   Quantification of (E). Shown are fold increases of Pearson correlation coefficients (P.C.C.) of LC3B and Gal3 signals, normalized to untreated siCtrl, representing LC3B colocalization with Gal3. Graph represents data from three independent experiments with ≥ 50 cells per condition.

G   Western blot analysis of lysates of cells transfected with UBE2QL1 or control siRNA for 72 h, treated with 200 nM Bafilomycin A1 for 5 h or 250 μM LLOMe for 3 h as indicated and probed with an antibody specific for LC3A/B. GAPDH was probed as loading control.

Data information: In (B, D, and F) data are presented as mean ± SD. *$P < 0.05$; ***$P < 0.001$ (one-way ANOVA with Bonferroni's multiple comparison test).
Source data are available online for this figure.

worms was largely aggravated with respect to number and size of Gal3-positive compartments compared to *scav-3(qx193)* alone (Fig 7G–I), indicating that UBC-25 is required to compensate the consequences of the loss of SCAV-3. Moreover, already the *ubc-25 (ok1732)* single mutant displayed increased lysosomal permeability compared to wild-type (N2) worms (Fig 7G–I) mirroring the occurrence of Gal3 puncta in UBE2QL1-depleted HeLa cells even without further challenge. Thus, UBE2QL1/UBC-25 has a conserved role in maintaining lysosomal integrity.

## Discussion

Extensive ubiquitination of lysosomal membrane proteins is a major hallmark of the response to lysosomal damage that is believed to mount the complex mechanisms leading to lysophagy of the damaged organelle. This work now identifies UBE2QL1 as a central and conserved ubiquitin-conjugating enzyme of the underlying ubiquitination machinery. UBE2QL1 acutely translocates to damaged lysosomes and associates with major regulators of the damage response. Crucially, UBE2QL1-mediated ubiquitination coordinates the recruitment of various critical effectors that are essential for autophagosome formation and lysophagy, and, consequently, for cell survival after damage. Thus, our data underscore the significance of ubiquitination in coordinating the critical steps of lysophagy and reveal a major player in the endolysosomal damage response with implications in cell homeostasis and degenerative disease.

Our results suggest that UBE2QL1 primarily regulates the formation of K48-linked ubiquitin chains because recruitment of the majority of UBE2QL1 coincides temporally and spatially with K48 chains at least judged by immunofluorescence with chain-specific antibodies. Moreover, depletion of UBE2QL1 blocks the extensive formation of K48-linked ubiquitin conjugates 2–3 h after lysosomal damage and the robust recruitment of the VCP/p97 AAA-ATPase, which primarily targets K48-conjugates [30]. These findings therefore directly establish the link between K48 chain formation and VCP/p97 recruitment in lysophagy, which is further supported by

the association of UBE2QL1 and VCP/p97 after lysosomal damage as detected by proximity biotinylation. So far, the targets and exact function of the UBE2QL1-K48-p97 axis are unclear. However, in analogy to the role of VCP/p97 in other processes such as mitophagy [25,31], it is likely that the pathway extracts proteins from the lysosomal membrane that interfere with p62/SQSTM1 polymerization or LC3 recruitment and thus hamper autophagosome formation [2,16]. These effects likely add up with the observed reduction of p62 on damaged lysosomes in UBE2QL1-depleted cells in compromising association of the LC3-decorated phagophore. We detect some UBE2QL1 on lysosomes as early as 30 min after damage indicating that UBE2QL1 could be also involved, directly or indirectly, in the early formation of K63-linked ubiquitin chains, and that this regulates p62 recruitment and polymerization. Alternatively, the effect on p62 could also be more directly attributed to UBE2QL1-generated K48 chains, because p62 recognizes these chains under certain conditions [32], and this concurs with the finding that K48 ubiquitination stimulates lysophagy [17].

An intriguing question is where on the damaged lysosome the UBE2QL1-mediated ubiquitination is initiated. By electron microscopy, we detected UBE2QL1 and K48 chains also in the lumen of damaged lysosomes. Moreover, galectins, which bind to luminal sugar chains after lysosomal damage, were strongly labeled in the UBE2QL1 proximity assay. This underscores the notion that sensing luminal features such as glycosylation is translated into a ubiquitination response that triggers lysophagy [12,16,17]. It also concurs with the finding that lysosomal transmembrane proteins, which we detect in proximity to UBE2QL1, are ubiquitinated on their luminal domains [17]. Of note, cellular depletion of Gal3 or Gal8 did not affect UBE2QL1 translocation to lysosomes suggesting an alternative recruitment mechanism. It will be important to clarify how luminal ubiquitination transduces signals to the cytosolic side of the damaged lysosome to promote lysophagy, in particular as the membrane permeabilization is thought to be transient [33] consistent with our ultrastructural analysis. So far, it is also unclear which ubiquitin ligase(s) cooperate(s) with UBE2QL1 in the pathway. The SCF subunit FBXO27, which mediates K48 ubiquitination during lysophagy, is not expressed in HeLa cells [17] that were analyzed in

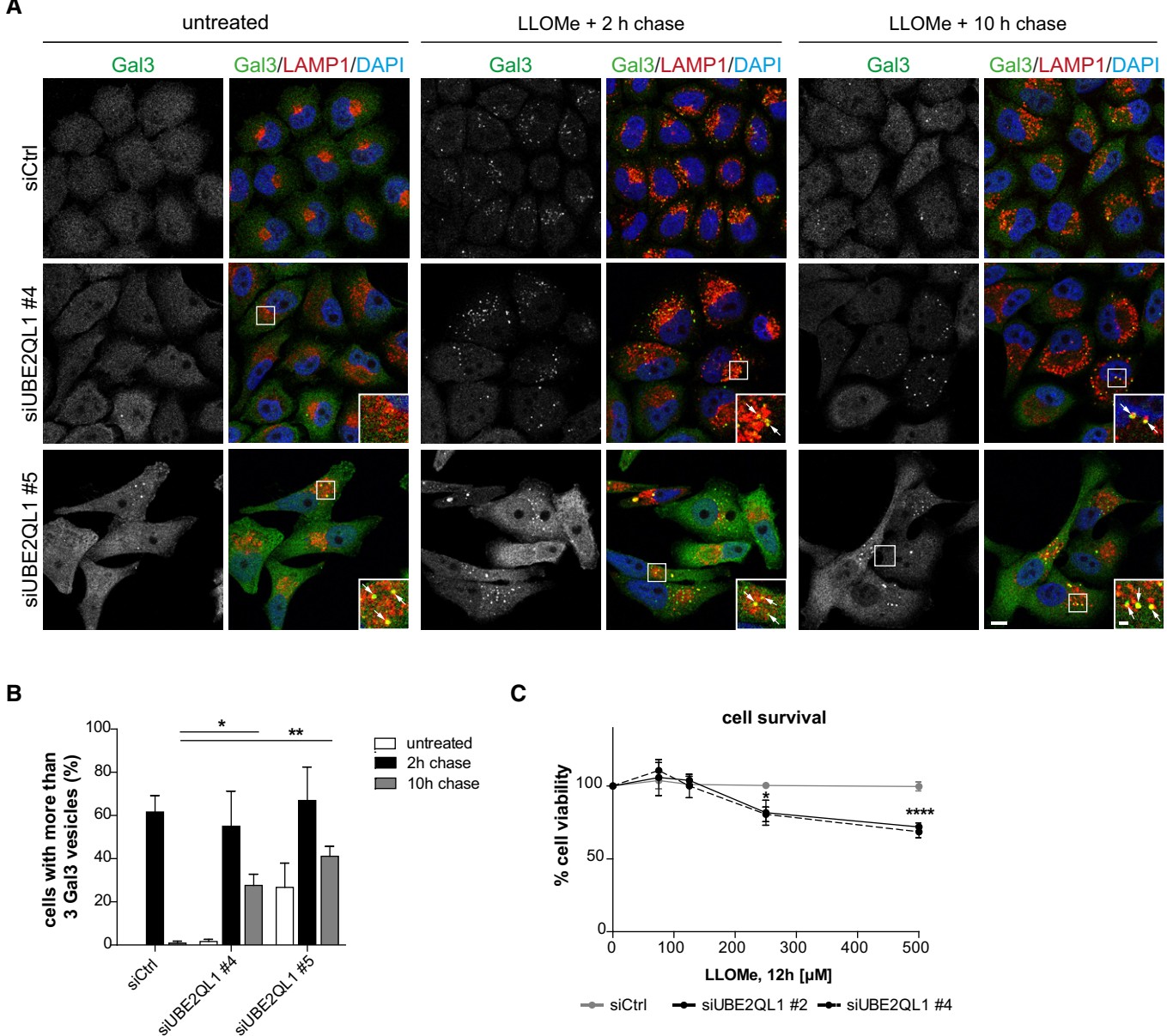

**Figure 6. UBE2QL1 is essential for efficient lysophagy and cell survival after lysosomal damage.**

A   Gal3 puncta assay for damaged lysosomes. HeLa cells were depleted of UBE2QL1 with two siRNAs for 60 h and treated with LLOMe or EtOH alone (untreated) for 1 h. Cells were fixed at indicated times after washout, stained with DAPI and Gal3 and LAMP1 antibodies, and processed for confocal microscopy. Note increased number of Gal3 puncta in untreated UBE2QL1-depleted cells and their persistence 10 h after LLOMe-induced damage. Scale bars: 10 μm, 2 μm for inlays. Arrows indicate Gal3 and LAMP1 colocalizing vesicles.

B   Automated quantification of (A). The percentage of cells with more than three Gal3 puncta is shown. Graph represents data from three independent experiments with ≥ 50 cells per condition (mean ± SD). *P < 0.05; **P < 0.01 (one-way ANOVA with Dunnett's multiple comparison test).

C   Survival assay. Control or UBE2QL1-depleted HeLa cells (48 h) were treated with increasing concentrations of LLOMe as indicated. Cell viability was measured with the MTS assay. Graph represents data from one experiment including three replicates (mean ± SD). *P < 0.05; ****P < 0.0001 (one-way ANOVA with Bonferroni's multiple comparison test).

this study. Moreover, TRIM16 has been reported to be critical for ubiquitination of damaged lysosomes [12], but it lacks a canonical E2-interaction domain. Thus, given the central role of UBE2QL1 for mounting the ubiquitination response on damaged lysosomes revealed in this study, its identification will help unravel further components of the machinery underlying the pathway.

Several pieces of evidence speak also for a constitutive role of UBE2QL1 in maintaining lysosome integrity. One is that TFEB is partially activated and translocated into the nucleus in UBE2QL1-depleted but otherwise unchallenged cells. Moreover, the number of lysosomes is increased and mTORC1 is dissociated from lysosomes after UBE2QL1 knockdown, although it cannot be excluded that this

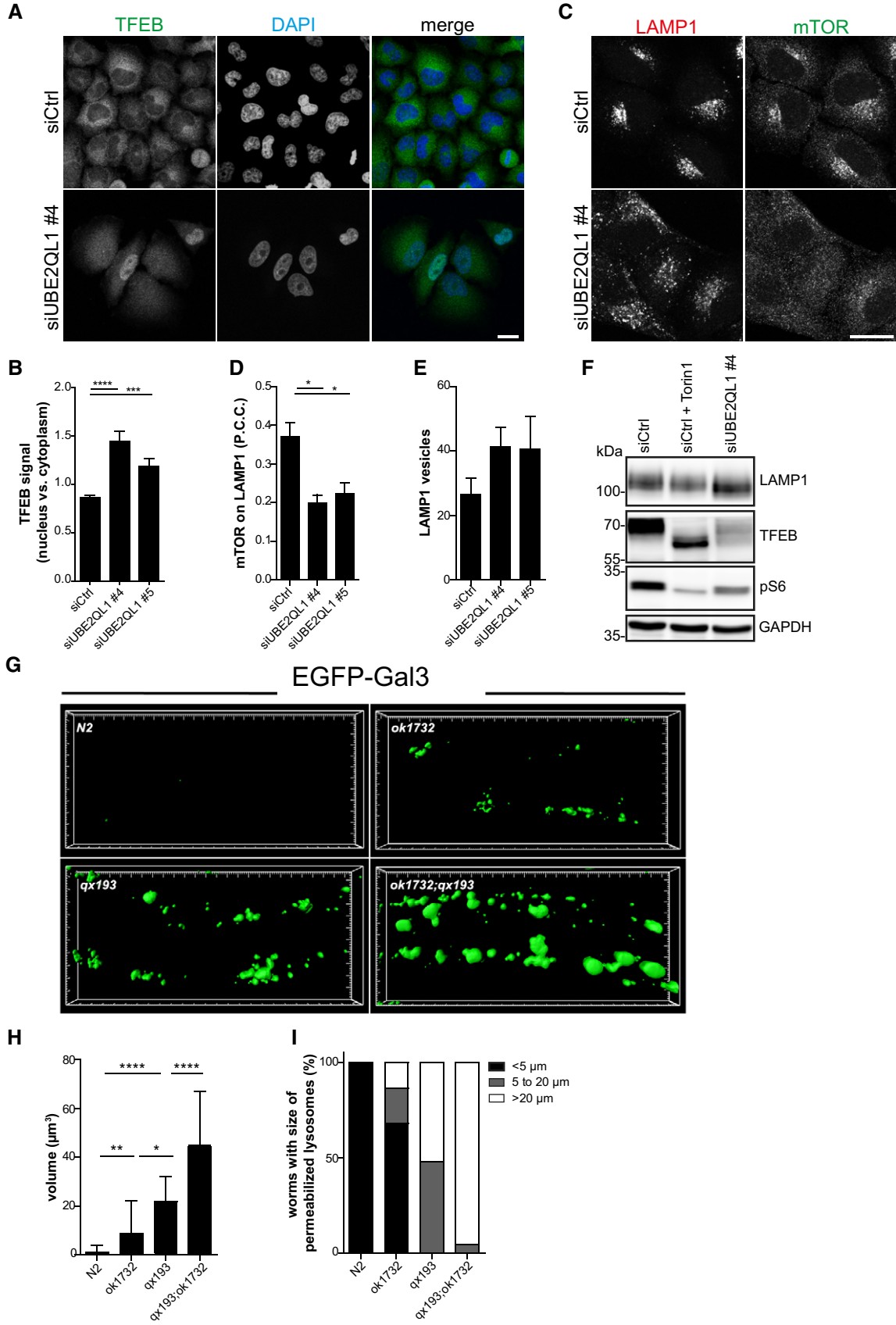

Figure 7.

**Figure 7. UBE2QL1 and its homologue UBC-25 in *Caenorhabditis elegans* are essential for maintenance of lysosomal integrity.**

A   Control or UBE2QL1-depleted (for 60 h) HeLa cells were stained with DAPI and an antibody against TFEB and automatically analyzed by confocal microscopy. Note nuclear translocation of TFEB upon loss of UBE2QL1. Scale bar: 20 µm. See Fig EV5A for images of cells treated with an alternative siRNA.

B   Automated quantification of (A). The intensity ratio of the cytoplasmic and nuclear TFEB signals was determined in both conditions. Graph represents data from four independent experiments with ≥ 100 cells per condition (mean ± SD). ***$P < 0.001$; ****$P < 0.0001$ (one-way ANOVA with Bonferroni's multiple comparison test).

C   Cells as in (A) were stained for LAMP1 and mTOR. Note the increase in lysosomes and lysosomal dissociation of mTOR in UBE2QL1-depleted cells. Scale bar: 20 µm. See Fig EV5B for images of cells treated with an alternative siRNA.

D   Automated quantification of (C). The colocalization of mTOR and LAMP1 is shown as Pearson correlation coefficients (P.C.C.) of the signals. Graph represents data from three independent experiments with ≥ 70 cells per condition (mean ± SD). *$P < 0.05$ (one-way ANOVA with Bonferroni's multiple comparison test).

E   Automated quantification of (C). Number of LAMP1-positive vesicles per cell is shown. Graph represents data from three independent experiments with ≥ 70 cells per condition (mean ± SD).

F   Western blot analysis of lysates of cells treated as in (C). As a positive control, cells were treated with the mTOR-inhibitor Torin1 (1 µM) for 2 h. GAPDH was probed as loading control.

G   *Caenorhabditis elegans* wild-type (N2) and mutant strains for *ubc-25(ok1732)*, for *scav-3(qx193)* or the *ubc-25* and *scav-3* double mutant (*ok1732;qx193*) carrying a EGFP-Gal3 transgene, were imaged in adult worms (L4 + 24 h). 3D reconstructions of the hypodermis. Small and big units represent 1 and 5 µm, respectively.

H   Average volume of permeabilized lysosomes within the region of hypodermis corresponding to 100 µm of body length was quantified. At least 20 animals were scored in each strain. *$P < 0.05$; **$P < 0.01$; ****$P < 0.0001$ (one-way ANOVA with Dunnett's multiple comparison test or Student's two-tailed unpaired *t*-test).

I   Populations of worms with different extent of lysosome permeabilization were analyzed. Graph represents percentage of worms with < 5, 5 to 20 or more than 20 permeabilized lysosomes in the region of hypodermis corresponding to 100 µm of body length.

Source data are available online for this figure.

is also due to yet unidentified functions of UBE2QL1. Notably, we find increased Gal3 puncta in UBE2QL1-depleted cells indicating that lysosome integrity is compromised in the absence of UBE2QL1. This may reflect a role distinct from lysophagy, because depletion of ATG5 and ATG7 in HeLa cells did not induce Gal3 puncta in unchallenged cells. The notion that UBE2QL1 regulates lysosomal integrity is further supported by the fact that mutation of the UBE2QL1 orthologue in *C. elegans*, ubc-25, causes increased lysosomal damage and exacerbates effects of the loss of SCAV-3/LIMP2. Thus, UBE2QL1 not only mediates the response to acute damage but also has a conserved role in maintaining lysosomal homeostasis. UBC-25 in worms is essential for the maintenance of the neuromuscular system [34]. Of note, mutations in VCP/p97 and p62/SQSTM1 cause muscular and neuronal degeneration in humans [35,36], and recent evidence demonstrates that compromised lysosome homeostasis is a main feature when VCP/p97 is inactivated [37]. This underlines the importance of the pathway including the associated ubiquitination machinery in lysosome integrity and in counteracting neuromuscular degeneration.

## Materials and Methods

### Plasmids

pmCherry-Gal3 and GFP-YOD1-C160S constructs were described previously [15]. Human UBE2J1 and human UBE2Q2 (transcript no. 1) cDNA were in pDNR223 (from ORFeome). Human UBE2QL1 cDNA (NM_001145161) with attB sites was synthesized (Integrated DNA Technologies, IDT) and cloned into pDNR221 using Gateway cloning (ThermoFisher Scientific). For localization screening, UBE2J1, UBE2Q2, and UBE2QL1 were cloned into pcDNA5FRT/TO-Strep/HA (N-terminal) [38] by Gateway cloning. For follow-up experiments, UBE2QL1 was cloned into pcDNA5FRT/TO-GFP (C-terminal) [39] and pcDNA5FRT/TO-Strep/HA (C-terminal) [40] using Gateway cloning. To generate the catalytically inactive form of UBE2QL1, site-directed mutagenesis was performed to obtain pcDNA5FRT/TO-Strep/HA (C-terminal) UBE2QL1 C88S. For

proximity proteomics, UBE2QL1 was cloned into pHAGE-C-Myc-APEX2 using Gateway cloning. pcDNA5FRT/TO-GFP p97 was cloned from pcDNA5/FRT/TO/myc-strep p97 [39] by site-directed mutagenesis to insert a *BamHI* restriction site and to exchange C-terminal myc with GFP tag.

### Cell lines

HeLa T-REx Flip-in cells (a kind gift of Gerhard Müller-Newen) were used to generate stable, doxycycline-inducible HeLa p97-GFP cells using pcDNA5/FRT/TO/GFP p97, as previously described [41]. HeLa mCherry-Gal3 cell line was generated as described [42]. Briefly, cells were transfected with linearized pmCherry-Gal3 plasmid, followed by selection in 600 µg/ml G418 for 3 days and 900 µg/ml G418 for 10 days, and were maintained in 500 µg/ml G418 in Dulbecco's modified Eagle's medium (DMEM) with 10% fetal bovine serum (FBS). HeLa cells stably expressing UBE2QL1-myc-APEX2 were obtained by lentiviral transduction followed by selection with antibiotics. Briefly, for virus generation, 1 µg of pMD2.G and 2.7 µg of pPAX2 retroviral packaging plasmid were transfected into 293T cells together with 3.3 µg of pHAGE-C-Myc-APEX2 UBE2QL1. Target HeLa cells were transduced with virus containing medium, which was exchanged to growth medium after 24 h. Transduced cells were selected with 2 µg/ml Puromycin (Sigma).

### Cell culture

HeLa cells were cultured in DMEM supplemented with 10% FBS in the presence of penicillin/streptomycin (pen/strep). Stable inducible p97-GFP HeLa FRT cells were cultured in DMEM supplemented with 10% tetracycline-free FBS in the presence of pen/strep with 250 µg/ml Hygromycin B (PAN-Biotech) and 15 µg/ml Blasticidin (ThermoFisher Scientific). Stable mCherry-Gal3 HeLa cells were cultured in DMEM supplemented with 10% FBS in the presence of pen/strep with 500 µg/ml G418 (VWR). Cells were transfected with plasmids using Lipofectamine 2000 (ThermoFisher Scientific) or with 10 nM siRNA (or 20 nM for the screen) using RNAiMAx (ThermoFisher

Scientific) according to the manufacturer's instructions. Transfected cells were analyzed after 24 h (plasmids) or 48, 60 or 72 h, as indicated (siRNA). For rescue experiments, cells were transfected with siRNAs for a total of 72 h. Medium was changed after 24 h, and cells were transfected with plasmids for another 48 h.

### RNA interference

siRNA targeting ATG5 and ATG7 was described before [15]. RNAi pools targeting E2 conjugating enzymes were purchased from Dharmacon. For the secondary screen, experiments were carried out using the four single siRNAs (ON-TARGET Plus, Dharmacon). For targeting UBE2QL1, siUBE2QL1 #5 (AAGCUGAAGCUACCUUU AATT) was from Microsynth; siUBE2QL1 #1 (J-024273-09) #2 (J-024273-10), #'3 (J-024273-11), and #4 (J-024273-12) were from Dharmacon. siGal3 (L-010606-00) was from Dharmacon. siGal8 #1 (CCCACGCCUGAAUAUUAAAGCAUUU) and siGal8 #2 (GGACAAAU UCCAGGUGGCUGUAAAU) [43] were from Microsynth. Non-coding siRNA was from Dharmacon (ON-TARGET plus, non-targeting pool, D-001810-10-05).

### Cell treatments

To induce lysosomal damage, HeLa cells were treated with 250 μM Leu-Leu methyl ester hydrobromide (LLOMe, Sigma) for indicated times or 8 μM Terfenadine (Sigma) for 24 h. Treatment with Alexa-488 labeled Tau fibrils was done as described before [15]. For depolarization of mitochondria, cells were treated with 10 μM Carbonyl cyanide 3-chlorophenylhydrazone (CCCP, Sigma-Aldrich) for 4 h. Cells were treated with 200 nM Bafilomycin A1 (Biomol) for 5 h or 1 μM Torin1 (Tocris Bioscience) for 2 h. Cell viability was measured using the 96$^{®}$ AQueous One Solution Cell Proliferation Assay (Promega).

### Antibodies

Rabbit anti-K48 ubiquitin chain (clone Apu2), rabbit anti-K63 ubiquitin chain (clone Apu3), and mouse anti-ubiquitin antibody (clone FK2) were purchased from Millipore. Mouse anti-LAMP1 (for IF) (sc-20011), rabbit anti-Tom20 (sc-11415), mouse anti-Gal3 (sc-32790) (for immuno-EM), and rabbit anti-Gal3 (sc-20157) (for IF) were purchased from Santa Cruz Biotechnology. Rabbit anti-p62 (P0067) (for IF), rabbit anti-LC3A/B (L8918) (for WB), mouse anti-GAPDH (G8795), rabbit anti-TAX1BP1 (HPA024432), and rabbit anti-HA (H6908) were from Sigma-Aldrich. Rabbit anti-mTOR (7C10), rabbit anti-TFEB (42405), rabbit anti-phospho S6 ribosomal protein (Ser235/236) (#2211), and rabbit anti-LAMP1 (C54H11) (for WB) were from Cell Signaling Technology. Goat anti-HA (ab9134) and rabbit anti-Gal3 (ab31707) (for WB) were from Abcam. Mouse anti-LC3B (LC3-1703) (for IF) was from Cosmo Bio. Mouse anti-p62 (610833) (for immuno-EM) was from BD Biosciences. Goat anti-GFP-Biotin (600-106-215) and rabbit anti-Biotin (100-4198) were from Rockland Immunochemicals. Goat anti-Gal8 (AF1305) was from R&D Systems. Rabbit anti-ATG5-ATG12 conjugate (NB110-53818) was from Novus Biologicals. Rabbit anti-ATG7 (AHP1651) was from AbD Serotec. Rabbit anti-UBE2QL1 was raised against the human C-terminal peptide (CTHEKYGWVTPPVSDG) and affinity purified. HRP-coupled

secondary antibodies were from Bio-Rad and Alexa Fluor-conjugated secondary antibodies from Invitrogen.

### Immunofluorescence staining and microscopy

Cells were fixed in 4% paraformaldehyde or 100% ice-cold methanol at −20°C (for TAX1BP1 staining), permeabilized with 0.1% Triton X-100 in PBS and blocked with 3% bovine serum albumin in PBS with 0.1% Triton X-100 and 0.1% saponin. Indirect immunofluorescence staining was followed by mounting in ProLong Gold (ThermoFisher Scientific). Confocal laser scanning microscopy was performed on a TCS SP5 AOBS system equipped with standard PMT detectors as well as sensitive HyD detectors, a 63×/1.4 NA oil immersion objective or an HC PL APO 20×/0.7NA dry objective. Lasers used were HeNe 633 nm, DPSS 561 nm, Ar 488 nm and Diode 405 nm. Acquisition and hardware was controlled by LAS AF software (Leica Microsystems).

Images were processed using Fiji software (https://imagej.net/ Fiji), Adobe Photoshop, and Illustrator. Automated quantifications were done with Cell Profiler software [44]. Graphs and statistical analysis were done using Excel (Microsoft Corporation) or GraphPad Prism (GraphPad Software).

For the siRNA screen, 384-well plates (PerkinElmer) were automatically imaged using an Opera automated spinning disk confocal microscope with a UPLAPO 60×/1.2NA water objective. Images were acquired with a High QE CCD camera and analyzed with the Acapella 2.6 Studio software (PerkinElmer).

### Immunoblotting

Cells were harvested 72 h after transfection with siRNAs and lysed in RIPA buffer composed of 25 mM Tris-HCl (pH 7.5), 150 mM NaCl, 1% Nonidet P-40, 1% Na-deoxycholate, and 0.1% SDS. For detection of LC3A/B, cells were lysed in lysis buffer consisting of 10 mM HEPES (pH 7.9), 10 mM KCl, 0.2 mM EDTA, 300 mM NaCl, and 1% Nonidet P-40. Both lysis buffers were supplemented with protease inhibitors (complete EDTA-free protease inhibitor cocktail, Roche) and phosphatase inhibitors (PhosStop, Roche). For detection of endogenous UBE2QL1, cells were harvested in lysis buffer of the lysosome enrichment kit (ThermoFisher Scientific #89839) and homogenized with a dounce tissue grinder. Samples were resolved by SDS–PAGE and transferred to nitrocellulose membranes (Amersham, GE Healthcare). Immunoblot analysis was performed with the indicated antibodies and visualized with SuperSignal West Pico Chemiluminescent substrate (Pierce).

### Immuno-electron microscopy

HeLa cells overexpressing UBE2QL1-GFP and HeLa cells co-overexpressing mCherry-Galectin-3 and GFP-YOD1-C160S were treated with 250 μM LLOMe for 3 h and then fixed for immuno-electron microscopy by adding freshly prepared 4% formaldehyde in 0.1 M phosphate buffer (pH 7.4) in the presence or absence of 0.2% glutaraldehyde (Polysciences Inc.) to an equal volume of culture medium. After 10 min, the fixative was refreshed to continue fixation for 2 h. Cells were stored in 1% formaldehyde at 4°C overnight. Ultrathin cryosectioning and immunogold labeling were performed as described [45]. The GFP constructs were detected by using goat

anti-GFP-Biotin and rabbit anti-Biotin. Galectin-3, p62, and K48 were detected by mouse anti-galectin-3, mouse anti-p62, and rabbit anti-K48, respectively. Primary antibodies were detected by protein A–10 nm or 15 nm gold particles (Cell Microscopy Center, Utrecht, The Netherlands).

### SILAC labeling and treatment

HeLa cells stably expressing UBE2QL1-APEX2 were grown in lysine- and arginine-free DMEM supplemented with FBS, L-glutamine, sodium pyruvate, heavy arginine (R10) (38 µg/ml) and lysine (K8) (66 µg/ml) or light arginine (R0) (38 µg/ml) and lysine (K0) (66 µg/ml), respectively. Further experiments were conducted as soon as the cells reached a protein labeling with heavy amino acids of at least 95%. Heavy-labeled cells were treated with 250 µM Leu-Leu methyl ester hydrobromide (LLOMe, Sigma) for 3 h at 37°C, while light-labeled cells were treated with vehicle alone (EtOH).

### Proximity labeling and mass spectrometry

Proximity labeling was performed in SILAC-labeled HeLa cells stably expressing UBE2QL1-APEX2 as described before [46]. Briefly, cells were incubated with 500 µM biotin phenol during the last 30 min of LLOMe treatment and subsequently pulsed by addition of $H_2O_2$ for 1 min at room temperature. To stop the biotinylation reaction, they were washed 3× with quencher solution (10 mM sodium azide, 10 mM sodium ascorbate, 5 mM Trolox in DPBS) and 3× with PBS. All further steps were performed at 4°C unless indicated otherwise. After cell harvest with 0.25% Trypsin/EDTA (ThermoFisher Scientific), cells were counted and heavy- and light-labeled cells were mixed at a 1:1 ratio based on total cell numbers. After centrifugation, the resulting cell pellets were lysed in APEX-RIPA (50 mM Tris, 150 mM NaCl, 0.1% SDS, 1% Triton X-100, 0.5% sodium deoxycholate) supplemented with 10 mM sodium ascorbate, 1 mM sodium azide, 1 mM Trolox, and protease inhibitors (Roche Complete). Samples were sonicated 2× for 1 s, spun down at $10,000 \times g$ for 10 min before application to streptavidin agarose resin (ThermoFisher Scientific), and incubation with overhead shaking overnight.

### Mass spectrometry

AP-MS with streptavidin was performed as described before [47]. Briefly, samples were washed 3× in APEX-RIPA buffer and 3× in 3 M urea buffer (in 50 mM ammonium bicarbonate) followed by incubation with TCEP (5 mM final) for 30 min at 55°C with shaking. After alkylation with IAA (10 mM final) for 20 min at room temperature in the dark, the reaction was quenched with DTT (20 mM final). Samples were washed 2× with 2 M urea (in 50 mM ammonium bicarbonate) before trypsin digestion overnight at 37°C (20 µg/ml final). The resin was spun down, and supernatants containing digested peptides were collected. After washing the resin 2× with 2 M urea and pooling all supernatants, the samples were acidified with TFA (1% final). Digested peptides were desalted on custom-made C18 stage tips. Using an Easy-nLC1200 liquid chromatography, peptides were loaded onto 75 µm × 15 cm fused silica capillaries (New Objective) packed with C18AQ resin (Reprosil-Pur 120, 1.9 µm, Dr. Maisch HPLC). Peptide mixtures were separated using a gradient of 5–33% acetonitrile in 0.5% acetic acid over

90 min and detected on an Orbitrap Elite mass spectrometer (ThermoFisher Scientific). Dynamic exclusion was enabled for 30 s, and singly charged species or species for which a charge could not be assigned were rejected. MS data were processed and analyzed using MaxQuant (version 1.6.0.1) [48] and Perseus (version 1.5.8.4; [49]). All proximity experiments were performed in quadruplicates. Unique and razor peptides were used for quantification. Matches to common contaminants, reverse identifications and identifications based only on site-specific modifications were removed prior to further analysis. Log2 H/L ratios were calculated. A threshold based on a log2 fold change of > 1.5-fold or < −1.5-fold was chosen so as to focus the data analysis on a small set of proteins with the largest alterations in abundance. Student's t-tests were used to determine statistical significance between treatments. A P-value < 0.05 was considered statistically significant.

### Caenorhabditis elegans

Strains of *C. elegans* were cultured and maintained using standard protocols [50]. The N2 Bristol strain was used as the wild-type strain. The strain RB1481 carrying allele *ubc-25 (ok1732)* is from CGC which is funded by NIH Office of Research Infrastructure Programs (P40 OD010440), and it was further outcrossed 4 times before phenotypic analysis. The strain carrying *scav-3(qx192)* was kindly provided by Prof. Xiaochen Wang. Transgenic animals carrying GFP-LGALS3 (EGFP-Gal3) were generated as described previously [10]. Stack images of the hypodermis were taken by confocal microscope LSM700 from Carl Zeiss. 3-dimensional reconstruction was done using the Bitplane software (Imaris).

## Data availability

The mass spectrometry proteomics data have been deposited to the ProteomeXchange Consortium (http://proteomecentral.proteomexchange.org) via the PRIDE [51] partner repository with the dataset identifier PXD014521.

Expanded View for this article is available online.

### Acknowledgements

HM was supported by the Deutsche Forschungsgemeinschaft (DFG) grants Me1626/5-1 and Me1626/4-2; CB was supported by DFG grants EXC2145 SyNergy and CRC1177, and by the Boehringer Ingelheim Foundation. MJ was supported by the Danish National Research Foundation (DNRF125) and the European Research Council (AdG 340751). We thank the Imaging Center Campus Essen (ICCE) for support with microscopy.

### Author contributions

LK designed and performed the majority of cell-based assays and immunoblots. CP performed cell-based assays. BK performed UBE2QL1 immunoblots and experiments with mitochondria. CP and BK helped with supervision of the project. GR helped with cell-based assays. TV and JK performed and analyzed EM. BL and MJ performed and analyzed *C. elegans* experiments. MP helped with the siRNA screen. AB and ME provided tau fibrils. LB performed the APEX2 experiments. CB supervised the screen and APEX experiments and analyzed data. HM conceived and supervised the project and wrote the paper. LK, BK, and CP helped with writing.

## Conflict of interest

The authors declare that they have no conflict of interest.

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
