## [Review Process File · EMBO Reports]

The ubiquitin-conjugating enzyme UBE2QL1 coordinates lysophagy in response to endolysosomal damage

Lisa Koerver, Chrisovalantis Papadopoulos, Bin Liu, Bojana Kravic, Giulia Rota, Lukas Brecht, Tineke Veenendaal, Mira Polajnar, Anika Bluemke, Michael Ehrmann, Judith Klumperman, Marja Jäättelä, Christian Behrends, Hemmo Meyer

Review timeline:

Submission date:	27 February 2018
Editorial Decision:	10 April 2018
Revision received:	1 July 2019
Editorial Decision:	31 July 2019
Revision received:	2 August 2019
Accepted:	7 August 2019

Editor: Martina Rembold/Achim Breiling

Transaction Report:

1st Editorial Decision

10 April 2019

Thank you for the submission of your research manuscript to our journal. I apologize for the delay in handling your manuscript but we have only now received the full set of referee reports that is copied below.

As you will see, the referees acknowledge that the findings are potentially interesting. However, the referees also point out several technical concerns and have a number of suggestions for how the study should be strengthened. In particular, both referee 1 and 3 are concerned that the different UBE2QL1 siRNA oligonucleotides have divergent effects and both indicate that further experiments to document the knockdown efficiency as well as rescue experiments are required. Ideally, these could be complemented with knockout cells, if these experiments are feasible.

Given these constructive comments, we would like to invite you to revise your manuscript with the understanding that the referee concerns (as detailed above and in their reports) must be fully addressed and their suggestions taken on board. Please address all referee concerns in a complete point-by-point response. Acceptance of the manuscript will depend on a positive outcome of a second round of review. It is EMBO reports policy to allow a single round of revision only and acceptance or rejection of the manuscript will therefore depend on the completeness of your responses included in the next, final version of the manuscript.

Revised manuscripts should be submitted within three months of a request for revision; they will otherwise be treated as new submissions. Please contact us if a 3-months time frame is not sufficient for the revisions so that we can discuss the revisions further.

Supplementary/additional data: The Expanded View format, which will be displayed in the main HTML of the paper in a collapsible format, has replaced the Supplementary information. You can submit up to 5 images as Expanded View. Please follow the nomenclature Figure EV1, Figure EV2

etc. The figure legend for these should be included in the main manuscript document file in a section called Expanded View Figure Legends after the main Figure Legends section. Additional Supplementary material should be supplied as a single pdf labeled Appendix. The Appendix includes a table of content on the first page with page numbers, all figures and their legends. Please follow the nomenclature Appendix Figure Sx throughout the text and also label the figures according to this nomenclature. For more details please refer to our guide to authors.

Regarding data quantification, please ensure to specify the name of the statistical test used to generate error bars and P values, the number (n) of independent experiments underlying each data point (not replicate measures of one sample), and the test used to calculate p-values in each figure legend. Discussion of statistical methodology can be reported in the materials and methods section, but figure legends should contain a basic description of n, P and the test applied. Please also include scale bars in all microscopy images.

We now strongly encourage the publication of original source data with the aim of making primary data more accessible and transparent to the reader. The source data will be published in a separate source data file online along with the accepted manuscript and will be linked to the relevant figure. If you would like to use this opportunity, please submit the source data (for example scans of entire gels or blots, data points of graphs in an excel sheet, additional images, etc.) of your key experiments together with the revised manuscript. Please include size markers for scans of entire gels, label the scans with figure and panel number, and send one PDF file per figure.

- a complete author checklist, which you can download from our author guidelines (<http://embor.embopress.org/authorguide#revision>). Please insert page numbers in the checklist to indicate where the requested information can be found.
 - a letter detailing your responses to the referee comments in Word format (.doc)
 - a Microsoft Word file (.doc) of the revised manuscript text
 - editable TIFF or EPS-formatted figure files in high resolution
- (In order to avoid delays later in the publication process please check our figure guidelines before preparing the figures for your manuscript:
http://www.embopress.org/sites/default/files/EMBOPress_Figure_Guidelines_061115.pdf)
- a separate PDF file of any Supplementary information (in its final format)
 - all corresponding authors are required to provide an ORCID ID for their name. Please find instructions on how to link your ORCID ID to your account in our manuscript tracking system in our Author guidelines (<http://embor.embopress.org/authorguide>).

As part of the EMBO publication's Transparent Editorial Process, EMBO reports publishes online a Review Process File to accompany accepted manuscripts. This File will be published in conjunction with your paper and will include the referee reports, your point-by-point response and all pertinent correspondence relating to the manuscript.

I look forward to seeing a revised version of your manuscript when it is ready. Please let me know if you have questions or comments regarding the revision.

REFeree REPORTS**Referee #1:**

The authors screened human E2 enzymes that are required for LLOMe-induced lysosomal ubiquitination and identified UBEQL1. UBEQL1 translocates to lysosomes upon lysosomal damage caused by LLOMe or tau fibrils. The kinetics of UBEQL1 recruitment correlates better with that of K48 ubiquitination than that of K63 ubiquitination. Immunoelectron microscopy and APEX2-based proximity biotinylation assay suggest that UBE2QL1 translocates to damaged lysosomes, probably into the lumen. Furthermore, UBEQL1 is required for recruitment of VCP, p62 and LC3 to damaged lysosomes and thereby for clearance of damaged lysosomes. Finally, the authors show the evidence that UBEQL1 plays constitutive roles such as regulation of TFEB and mTORC1 and maintenance of lysosomal membrane integrity.

This study was carefully performed and the results are convincing. The discovery of the E2 enzyme UBEQL1 and demonstration of its role upon lysosomal damage as well as under normal conditions are important and informative to the field. Remaining questions would be how this E2 enzyme translocates to damaged lysosomes and what E3 ligase(s) cooperate with UBEQL1, but these could be beyond the scope of this initial study. This study would be strengthened by addressing the following points.

Major comments:

1. This study uses only knockdown cells, which results in some inconsistencies. The efficiency of the four siRNAs against UBEQL1 is different in the screen: #4 displays the highest effect, but #1 and #3 show no effect. Does this result correlate with their knockdown efficiency? Knockdown efficiencies of these four siRNA should be shown. Also, the knockdown efficiency is not ideal (Fig. 1G) and Gal3 vesicles are observed only in the untreated siUBE2QL1 M1 cells and not in the untreated the D4 cells when quantified (Fig. 6B). Do M1, D2, and D4 correspond to #1, #2, and #4 in the screen? The use of knockout cells is more ideal, but the authors failed to generate it. Is UBE2QL1 indeed essential in HeLa cells? According to the paper by Wang et al. (Science. 2015, 350:1096-101), essentiality of UBE2QL1 seems to be not high. Since *C. elegans* UBC-25 mutant is viable, could the authors generate knockout cells as well?
2. Although the authors use two independent siRNAs in some experiments, this reviewer thinks that rescue experiments using siRNA-resistant constructs (wild-type and catalytic mutant) is essential in key experiments such as in Figs. 2, 6, and 7.
3. In Fig. 6A, are the Gal3 puncta in untreated cells indeed lysosomes? To suggest a role of UBE2QL1 under basal conditions, counterstaining with lysosomal markers is required.
4. The authors show that knockdown of UBE2QL1 results in delayed clearance of Gal3+ lysosomes, reduced viability of LLOMe-treated cells, accumulation of Gal3+ lysosomes and activation of TFEB even without LLOMe treatment. The authors discussed that these are caused by a defect in lysophagy. If so, knockdown of ATG genes such as FIP200 and ATG7 should exhibit a similar phenotype. This experiment would be important to determine whether the phenotype observed is dependent on autophagy or not.
5. While the precise mechanism of UBEQL1 recruitment may be beyond the scope of this manuscript, the authors should at least test whether galectins 3/8 are required.

Minor comments:

1. As the authors show in Fig 3A and B, the timing of p62 recruitment does not seem to be dependent on UBE2QL1 activity. However, p62 recruitment to damaged lysosomes in UBE2QL1-depleted cells is reduced (Fig 5C and D). Can the authors reconcile these 2 pieces of data?
2. It has been suggested that UBE2QL1 is important for cell growth and mTOR activity (Wake et

al., Hum Mutat 2013). It should be discussed that UBEQL1 may not be specific to lysophagy.

3. The images of TFEB staining (Fig. 7A and EV7A) are not of good quality. Many of the nuclei appear to be out of focus, making the evaluation of nuclear translocation difficult.

Referee #2:

Koerver and co-workers have used an siRNA screening protocol for identification of ubiquitin E2-conjugating enzymes involved in ubiquitination of damaged lysosomes. UBE2QL1 was identified as a hit in this screen, and microscopy experiments revealed that both exogenous and endogenous UBE2QL1 is recruited to lysosomes upon LLOMe-induced damage. UBE2QL1 was found to mediate mainly K48-linked polyubiquitination, and siRNA-mediated knockdown prevented recruitment of VCP/p97 and p62 to damaged lysosomes. Consistent with this, recruitment of the autophagy marker LC3B was also reduced by UBE2QL1, indicating that UBE2QL1-mediated ubiquitination of lysosomal substrates is important for lysophagy. Depletion of UBE2QL1 in cells not treated with LLOMe caused increased lysosome damage (as revealed with Galectin 3), mTORC1 dissociation from lysosomes, and activation of TFEB. Lysosome destabilization was also detected in *C.elegans* lacking SCAV-3/LIMP2 in addition to UBE2QL1. The authors conclude that UBE2L1 is a critical coordinator of the acute endolysosomal damage response, thereby ensuring lysosome integrity.

Overall these are very interesting results, even in the absence of any molecular mechanism of UBE2QL1 activation by lysosome damage, and information about E3 ubiquitin ligases functioning downstream of UBE2QL1 in the endolysosomal damage response. The study has been performed in a very competent manner, with high-quality molecular biology, fluorescence and electron microscopy experiments, and with adequate quantifications. I thus think this manuscript is a good fit for EMBO Reports and have only very minor comments:

Minor point:

From the results in Figure 7, the authors conclude that "UBE2QL1 and its homologue UBC-25 in *C. elegans* are essential for maintenance of lysosomal integrity." This is an overinterpretation of the data since the authors have strictly monitored nuclear translocation of TFEB. Surely, there are multiple cellular events in addition to lysosome damage that can affect TFEB localization.

Referee #3:

The presented study by Meyer and colleagues investigated the ubiquitination machinery necessary for lysosomal ubiquitination upon damage and subsequent lysophagy. They focused on E2 ubiquitin conjugating enzymes and employed siRNA- and image-based screens to identify essential E2 enzymes for LLOMe-induced ubiquitination of lysosomal proteins. The strongest candidate was UBE2QL1, which was then characterized in more detail. Furthermore, the authors showed that UBE2QL1 is necessary for the recruitment of key regulators of lysophagy like p97 and p62, and for subsequent organelle clearance and cell survival after lysosomal damage. Using *C. elegans* as a model organism, they also provide evidence that UBE2QL1 has a conserved role on maintaining lysosomal integrity.

Overall, the presented study describes a very interesting, new and topical finding that is of interest to the broad readership of EMBO reports. The quality of the work is high, complementary approaches were used to establish UBE2QL1 as important regulator of the acute endolysosomal damage response and for maintenance of lysosomal integrity. I recommend publication after addressing the following points:

- Figure 1: Only one out of four individual UBE2QL1 siRNA oligonucleotides caused a significant difference in numbers of FK2 positive (or K48 positive) LAMP1 vesicles (Fig. 1D). The authors claim that this is due to different knockdown efficiencies. Since no complementary knockout method was successful in HeLa cells, the authors need to expand Fig. 1G and include all siRNAs that they used to substantiate their statement.

Most of the following experiments in the manuscript included two different siRNAs, which reduced

the risk that the phenotype of siRNA #4 is due to off-target effects. However, for none of these experiments knockdown efficiency was proven (e.g. by Western blot or IF).

- Figure 2/3: In general, the authors should use more careful wording regarding the ubiquitin linkage type mediated by UBE2QL1. In my opinion, they should remove "is essential" from the following sentence: "Thus, whereas UBE2QL1-depletion also affects K63 ubiquitination, the bulk of UBE2QL1 coincides with and is essential for the late K48-linked ubiquitination response."

First, it is known that the Ub linkage type-specific antibodies are not always reliable (there are huge batch-to-batch variations) and one needs to be careful with conclusions drawn from these experiments, unless additional controls are included (e.g. knockdown of linkage-specific E2 enzyme).

If the authors want to make a strong statement about linkage types, they could for example isolate lysosomes from control and UBE2QL1-depleted cells after LLOMe-treatment and quantify Ub linkage types by MS. Alternatively, they could immunoprecipitate LAMP1 or LAMP2 and analyse by MS. Without these additional controls, the authors should be more conservative in describing linkage-specificity of UBE2QL1 throughout the manuscript.

1st Revision - authors' response

1 July 2019

Point-by-Point response

Referee #1:

"The authors screened human E2 enzymes that are required for LLOMe-induced lysosomal ubiquitination and identified UBEQL1. UBEQL1 translocates to lysosomes upon lysosomal damage caused by LLOMe or tau fibrils. The kinetics of UBEQL1 recruitment correlates better with that of K48 ubiquitination than that of K63 ubiquitination. Immunoelectron microscopy and APEX2-based proximity biotinylation assay suggest that UBE2QL1 translocates to damaged lysosomes, probably into the lumen. Furthermore, UBEQL1 is required for recruitment of VCP, p62 and LC3 to damaged lysosomes and thereby for clearance of damaged lysosomes. Finally, the authors show the evidence that UBEQL1 plays constitutive roles such as regulation of TFEB and mTORC1 and maintenance of lysosomal membrane integrity.

This study was carefully performed and the results are convincing. The discovery of the E2 enzyme UBEQL1 and demonstration of its role upon lysosomal damage as well as under normal conditions are important and informative to the field. Remaining questions would be how this E2 enzyme translocates to damages lysosomes and what E3 ligase(s) cooperate with UBEQL1, but these could be beyond the scope of this initial study. This study would be strengthened by addressing the following points. "

Major comments:

"1. This study uses only knockdown cells, which results in some inconsistencies. The efficiency of the four siRNAs against UBEQL1 is different in the screen: #4 displays the highest effect, but #1 and #3 show no effect. Does this result correlate with their knockdown efficiency? Knockdown efficiencies of these four siRNA should be shown. Also, the knockdown efficiency is not ideal (Fig. 1G) and Gal3 vesicles are observed only in the untreated siUBE2QL1 M1 cells and not in the untreated the D4 cells when quantified (Fig. 6B). Do M1, D2, and D4 correspond to #1, #2, and #4 in the screen?..."

We apologize if the numbering of the siRNA was confusing and if this made it difficult to follow that the results are in fact consistent. We changed the numbering to #1-4 (the commercial oligos from the screen, formerly D1-4) and #5 (formerly M1). We designed #5 ourselves to target the open reading frame while the commercial oligos target UTRs. As requested, we performed additional Western blots to probe the knockdown efficiency for #1 and #3 (that did not score in the screen). It is now shown in Appendix Fig. S1. As expected, and consistent with the results from the screen, oligos #1 and #3 deplete less efficiently if at all. This is not unusual for a commercial library. Please note that the gene is small and GC-rich. The design of efficient siRNA oligos is therefore difficult.

The referee states that the “knockdown efficiency is not ideal”. We would argue that it is difficult to predict from Western blots at what degree of knockdown the protein function is compromised. We showed the knockdown efficiency for the relevant oligos #2,#4,#5 also by immunofluorescence. Fig. EV1 B and C clearly shows the significant reduction of UBE2QL1 on damaged lysosomes, which is the relevant pool for its activity in lysophagy.

As for the Gal3 puncta in unchallenged cells, we also see them in depletions with oligo #4 when depletions are extended to 72h (rather than 60h for #5 in Fig. 6). These data are now shown in Appendix Fig. S3A.

We hope it now becomes clear that our siRNA results are not inconsistent. On the contrary, the degree of effects corresponds to the knockdown efficiencies of the siRNA oligos. The requested rescue experiments (see point 2) will hopefully further clarify this issue.

“(point 1 continued)...The use of knockout cells is more ideal, but the authors failed to generate it. Is UBE2QL1 indeed essential in HeLa cells? According to the paper by Wang et al. (Science. 2015, 350:1096-101), essentiality of UBE2QL1 seems to be not high. Since C. elegans UBC-25 mutant is viable, could the authors generate knockout cells as well? “

As stated in the text, we did not manage to generate HeLa UBE2QL1 knockout cells by CRISPR/CAS technology (whereas this worked for many other genes in unrelated projects in the lab). We tried a commercial set of sgRNAs and also used self-designed sgRNAs. For this revision, we have in the meantime also tried the knockout in RCC4 and RPE-1 cells, but again failed. The fact that we could not isolate knockout cells is consistent with siRNA-mediated knockdown of UBE2QL1 causing a growth delay in HeLa and in fact also with the problems of the C.elegans mutant. While many E2s are not essential in the worm, the *ubc-25(ok1732)* mutant has reduced self-brood size and increased embryonic lethality (Roy et al, 2014; doi: 10.1534/g3.114.010546), indicating that UBC-25 is important for survival.

“2. Although the authors use two independent siRNAs in some experiments, this reviewer thinks that rescue experiments using siRNA-resistant constructs (wild-type and catalytic mutant) is essential in key experiments such as in Figs. 2, 6, and 7.”

As requested, we now provide rescue experiments to prove the specificity of UBE2QL1 depletion (Fig. EV2). We chose siRNA #4 because it targets the UTR. The overexpressed cDNA is therefore resistant to the siRNA. Please note that these experiments were very laborious, because the conditions had to be optimized in particular in terms of timing of the treatments. This is because UBE2QL1-depleted cells are difficult to transfect most likely due to the effect on the endolysosomal system. Moreover, the additional cDNA transfection impose extra stress on the cells.

Nevertheless, in Fig. EV2, we now clearly show that UBE2QL1 wt, but not the catalytically-inactive C88S mutant rescues ubiquitination of damaged lysosomes in UBE2QL1-depleted cells. Note that this was evaluated in biological replica with automated image quantification. Knockdown and overexpression was monitored by Western blot analysis (Fig. EV2C). In Fig. EV5C, we also show that overexpression of UBE2QL1 wt, but not C88S, partially rescues the effect on TFEB, S6 phosphorylation and the LAMP1 increase in untreated cells. Thus, we have rescued two different aspects of the UBE2QL1 knockdown effect which further strengthens our conclusion.

“3. In Fig. 6A, are the Gal3 puncta in untreated cells indeed lysosomes? To suggest a role of UBE2QL1 under basal conditions, counterstaining with lysosomal markers is required.”

As requested, we now provide the counterstaining (Fig. 6A and S3A). In general, the Gal3 puncta very faithfully colocalize with the lysosomal markers.

“4. The authors show that knockdown of UBE2QL1 results in delayed clearance of Gal3+ lysosomes, reduced viability of LLOMe-treated cells, accumulation of Gal3+ lysosomes and activation of TFEB even without LLOMe treatment. The authors discussed that these are caused by a defect in lysophagy. If so, knockdown of ATG genes such as FIP200 and ATG7 should exhibit a similar phenotype. This experiment would be important to determine whether the phenotype

observed is dependent on autophagy or not.”

We performed this experiment for ATG5 and ATG7 depletion, and saw Gal3 puncta only in very few cells (Fig. S3B), suggesting that the role of UBE2QL1 in lysosome homeostasis (of unchallenged cells) is independent of lysophagy. This is in line with most recent data in mice showing that p97 inactivation has a more pronounced effect on lysosome integrity of muscle than inactivation of autophagy genes (Arhzaouy et al., 2019). We now also discuss this issue in the discussion section.

“5. While the precise mechanism of UBE2QL1 recruitment may be beyond the scope of this manuscript, the authors should at least test whether galectins 3/8 are required.”

As requested, we now provide these experiments. Knockdown of Gal3 and Gal8 was efficient, but (either alone or in combination) had no effect on UBE2QL1 translocation to damaged lysosomes, arguing that recruitment of UBE2QL1 is independent of these galectins. The data is now included in Appendix Fig. S2A-C and discussed in the text.

Minor comments:

“1. As the authors show in Fig 3A and B, the timing of p62 recruitment does not seem to be dependent on UBE2QL1 activity. However, p62 recruitment to damaged lysosomes in UBE2QL1-depleted cells is reduced (Fig 5C and D). Can the authors reconcile these 2 pieces of data?”

We demonstrate that some p62 is recruited earlier than UBE2QL1, but that its recruitment is consolidated by further ubiquitination that is mediated by UBE2QL1. This may have to do with its ability to also bind K48 ubiquitin chains and to polymerize, which may only be triggered by additional ubiquitination mediated by UBE2QL1. We discussed this in the discussion section.

“2. It has been suggested that UBE2QL1 is important for cell growth and mTOR activity (Wake et al., Hum Mutat 2013). It should be discussed that UBE2QL1 may not be specific to lysophagy.”

We agree. We now mention this possibility in the discussion section (last paragraph).

“3. The images of TFEB staining (Fig. 7A and EV7A) are not of good quality. Many of the nuclei appear to be out of focus, making the evaluation of nuclear translocation difficult.”

All images are in focus. This becomes clear in the DAPI channel that we now provide separately in the figure (Fig. 7A). The reviewer’s impression is probably caused by the fact that in some cells, TFEB localizes to the cytoplasm and the nucleus, while in others translocation (and therefore the nuclear shape) is more pronounced. In contrast, control cells do not display this partial translocation and the majority of TFEB is in the cytoplasm. Note that this is confirmed by automated image quantification (ratio nucleus/cytoplasm in four biological replica).

Referee #2:

“Koerver and co-workers have used an siRNA screening protocol for identification of ubiquitin E2-conjugating enzymes involved in ubiquitination of damaged lysosomes. UBE2QL1 was identified as a hit in this screen, and microscopy experiments revealed that both exogenous and endogenous UBE2QL1 is recruited to lysosomes upon LLOMe-induced damage. UBE2QL1 was found to mediate mainly K48-linked polyubiquitination, and siRNA-mediated knockdown prevented recruitment of VCP/p97 and p62 to damaged lysosomes. Consistent with this, recruitment of the autophagy marker LC3B was also reduced by UBE2QL1, indicating that UBE2QL1-mediated ubiquitination of lysosomal substrates is important for lysophagy. Depletion of UBE2QL1 in cells not treated with LLOMe caused increased lysosome damage (as revealed with Galectin 3), mTORC1 dissociation from lysosomes, and activation of TFEB. Lysosome destabilization was also detected in C.elegans lacking SCAV-3/LIMP2 in addition to UBE2QL1. The authors conclude that UBE2QL1 is a critical coordinator of the acute endolysosomal damage response, thereby ensuring lysosome integrity.”

Overall these are very interesting results, even in the absence of any molecular mechanism of UBE2QL1 activation by lysosome damage, and information about E3 ubiquitin ligases functioning downstream of UBE2QL1 in the endolysosomal damage response. The study has been performed in a very competent manner, with high-quality molecular biology, fluorescence and electron microscopy experiments, and with adequate quantifications. I thus think this manuscript is a good fit for EMBO Reports and have only very minor comments:

Minor point:

From the results in Figure 7, the authors conclude that "UBE2QL1 and its homologue UBC-25 in C. elegans are essential for maintenance of lysosomal integrity." This is an overinterpretation of the data since the authors have strictly monitored nuclear translocation of TFEB. Surely, there are multiple cellular events in addition to lysosome damage that can affect TFEB localization. "

We agree and thank the referee for this comment. Please note that our statement on lysosomal integrity at this point refers to many observations including the Gal3 signals in HeLa and worms. Nevertheless, we now discuss the possibility of other effects on TFEB in the discussion section.

Referee #3:

"The presented study by Meyer and colleagues investigated the ubiquitination machinery necessary for lysosomal ubiquitination upon damage and subsequent lysophagy. They focused on E2 ubiquitin conjugating enzymes and employed siRNA- and image-based screens to identify essential E2 enzymes for LLOMe-induced ubiquitination of lysosomal proteins. The strongest candidate was UBE2QL1, which was then characterized in more detail. Furthermore, the authors showed that UBE2QL1 is necessary for the recruitment of key regulators of lysophagy like p97 and p62, and for subsequent organelle clearance and cell survival after lysosomal damage. Using C. elegans as a model organism, they also provide evidence that UBE2QL1 has a conserved role on maintaining lysosomal integrity.

Overall, the presented study describes a very interesting, new and topical finding that is of interest to the broad readership of EMBO reports. The quality of the work is high, complementary approaches were used to establish UBE2QL1 as important regulator of the acute endolysosomal damage response and for maintenance of lysosomal integrity. I recommend publication after addressing the following points: "

"- Figure 1: Only one out of four individual UBE2QL1 siRNA oligonucleotides caused a significant difference in numbers of FK2 positive (or K48 positive) LAMP1 vesicles (Fig. 1D). The authors claim that this is due to different knockdown efficiencies. Since no complementary knockout method was successful in HeLa cells, the authors need to expand Fig. 1G and include all siRNAs that they used to substantiate their statement.

Most of the following experiments in the manuscript included two different siRNAs, which reduced the risk that the phenotype of siRNA #4 is due to off-target effects. However, for none of these experiments knockdown efficiency was proven (e.g. by Western blot or IF). "

As requested, we now provide a Western blot showing the knockdown efficiency for all the siRNA oligos used in the study (see new Appendix Fig. S1A). Please note that, for clarity, we changed the numbering to #1-4 (from the screen, formerly D1-4) and #5 (formerly M1). As expected, oligos #1 and #3 depleted UBE2QL1 less efficiently if at all, consistent with the fact that they did not score in the screen. The knockdown efficiency of the other oligos correspond to the degree of effects in our assays. While the knockdown is not complete according to the Western blot, we showed by IF that it significantly reduced the amount of UBE2QL1 recruited to damaged lysosomes (Fig. EV1B). It is true that we do not confirm the knockdown in each experiment, but the depletion is consistent and experiments were done under comparable conditions. Moreover, the consistent penetrance of treatments was visible by the effect on lysosomes (as seen in Fig. 7C) in each experiment.

"- Figure 2/3: In general, the authors should use more careful wording regarding the ubiquitin linkage type mediated by UBE2QL1. In my opinion, they should remove "is essential" from the following sentence: "Thus, whereas UBE2QL1-depletion also affects K63 ubiquitination, the bulk of

UBE2QL1 coincides with and is essential for the late K48-linked ubiquitination response." First, it is known that the Ub linkage type-specific antibodies are not always reliable (there are huge batch-to-batch variations) and one needs to be careful with conclusions drawn from these experiments, unless additional controls are included (e.g. knockdown of linkage-specific E2 enzyme).

If the authors want to make a strong statement about linkage types, they could for example isolate lysosomes from control and UBE2QL1-depleted cells after LLOMe-treatment and quantify Ub linkage types by MS. Alternatively, they could immunoprecipitate LAMP1 or LAMP2 and analyse by MS. Without these additional controls, the authors should be more conservative in describing linkage-specificity of UBE2QL1 throughout the manuscript. "

As requested, we made the respective text change and discuss the issue more cautiously at various places. We also removed the reference to K48 from the abstract. Although the temporal distribution of the respective signals (K63 vs K48 antibodies) is very consistent, we agree that this issue requires more clarification. This is also not essential for our conclusion.

2nd Editorial Decision

31 July 2019

Thank you for the submission of your revised manuscript to EMBO reports. It has been sent back to former referee #1 and #3 and we have now received their reports that are copied below.

As you will see, both referees are very positive about the study and support its publication in EMBO reports without further revision.

Browsing through the manuscript myself, I noticed a few editorial things that we need before we can proceed with the official acceptance of your study.

- 1) Our data editors from Wiley have already inspected the Figure legends for completeness and accuracy. Please see their suggested changes in the attached Word file. I have also taken the liberty to make some changes to the Abstract (mainly to convert it to present tense). Could you please review it?
- 2) Please note that the author checklist is published along with the review process file. I therefore suggest updating section 'F' on Data availability.
- 3) Please provide page numbers for the Appendix and add them to the Appendix table of content on the first page.
- 4) I wonder whether the source data for Figure 4D-F should not be uploaded as Dataset EV1. In case you think that these data are valuable as Dataset, please rename it to Dataset EV1 and provide a legend for it, either within the first row of the Excel file or in a separate tab.
- 5) Source data for Figure EV5C: the position of the MW marker for TFEB does not correspond between the source data file and the figure panel. Please double-check.
- 6) Figure callouts in the text: We noticed that Fig EV3C is not called out in the main text. Moreover, there is a callout for Fig EV4C on page 7 but there is no such figure panel. Please double-check this reference.
- 7) Figure EV5C: you mention that the blots for UBEQL1 are identical to Fig EV2C and are shown for clarity. The blot for GAPDH is also identical to Fig EV2C, which should be mentioned as well. Moreover, are all these blots from the same experiment/lysate, i.e., also the UBE2QL1 and the GAPDH blot, which serves here as control blot for pS6? Please clarify.
- 8) Finally, EMBO Press is pleased to support the "minimum reporting standards in the life sciences" initiative (<https://osf.io/preprints/metaarxiv/9sm4x/>). This effort brings together a number of leading journals and reproducibility experts to develop minimum expectations for reporting information

about Materials (including data and code), Design, Analysis and Reporting (MDAR) in published papers. We believe broad alignment on these standards will be to the benefit of authors, reviewers, journals and the wider research community and will help drive better practise in publishing reproducible research.

We are therefore participating in a community pilot involving a small number of life science journals to test the MDAR checklist. The checklist is intended to help authors, reviewers and editors adopt and implement the minimum reporting framework.

We very much hope that you will be willing to participate in this trial; the MDAR reporting checklist and an MDAR elaboration document providing context for the standards is attached to this message. If you agree to participate, please complete the MDAR reporting checklist and return it to us within 7 days. We would also be very grateful if you could complete this author survey <https://forms.gle/FRx7hpKS8g1QMNP9>.

Please note that your completed checklist and survey will be shared with the minimum reporting standards working group. However, the working group will not be provided with access to the manuscript or any other confidential information including author identities, manuscript titles or abstracts. Feedback from this process will be used to consider next steps, which might include revisions to the content of the checklist. Data and materials from this initial trial will be publicly shared in September 2019. Data will only be provided in aggregate form and will not be parsed by individual article or by journal, so as to respect the confidentiality of responses.

Please treat the checklist and elaboration as confidential as public release is planned for September 2019.

If you decide against participating, we would be grateful for any feedback you may have.

REFEREE REPORTS

Referee #1:

All the points raised on the first version of the MS have been properly addressed by the authors.

Referee #3:

Meyer and colleagues convincingly addressed all comments and concerns raised during the review process. I recommend publication of their manuscript "The ubiquitin-conjugating enzyme UBE2QL1 coordinates lysophagy in response to endolysosomal damage" now without further delay.

2nd Revision - authors' response

2 August 2019

The authors performed all minor editorial changes.

Corresponding Author Name: Hemmo Meyer

Journal Submitted to: EMBO reports

Manuscript Number: